# Mitigating memory effects during undulatory locomotion on hysteretic materials

**Perrin E Schiebel[1]\***, **Henry C Astley[1,2]**, **Jennifer M Rieser[1]**, **Shashank Agarwal[3]**, **Christian Hubicki[1,4]**, **Alex M Hubbard[1]**, **Kelimar Diaz[1]**, **Joseph R Mendelson III[5,6]**, **Ken Kamrin[4]**, **Daniel I Goldman[1]\***

[1]Department of Physics, Georgia Institute of Technology, Atlanta, United States; [2]Biology and the Department of Polymer Science, University of Akron, Akron, United States; [3]Department of Mechanical Engineering, Massachusetts Institute of Technology, Cambridge, United States; [4]Department of Mechanical Engineering, Florida A&M University-Florida State University, Tallahassee, United States; [5]School of Biological Sciences, Georgia Institute of Technology, Atlanta, United States; [6]Zoo Atlanta, Atlanta, United States

**Abstract** While terrestrial locomotors often contend with permanently deformable substrates like sand, soil, and mud, principles of motion on such materials are lacking. We study the desert-specialist shovel-nosed snake traversing a model sand and find body inertia is negligible despite rapid transit and speed dependent granular reaction forces. New surface resistive force theory (RFT) calculation reveals how wave shape in these snakes minimizes material memory effects and optimizes escape performance given physiological power limitations. RFT explains the morphology and waveform-dependent performance of a diversity of non-sand-specialist snakes but overestimates the capability of those snakes which suffer high lateral slipping of the body. Robophysical experiments recapitulate aspects of these failure-prone snakes and elucidate how re-encountering previously deformed material hinders performance. This study reveals how memory effects stymied the locomotion of a diversity of snakes in our previous studies (Marvi et al., 2014) and indicates avenues to improve all-terrain robots.

**\*For correspondence:**
perrin.schiebel@gatech.edu (PES);
daniel.goldman@physics.gatech.edu (DIG)

**Competing interests:** The authors declare that no competing interests exist.

## Introduction

Movement is crucial to the survival of most organisms and a necessary ability in robots used in fields like medicine (*Taylor, 2006*), search and rescue (*Murphy et al., 2008*), and extraterrestrial exploration (*Lindemann and Voorhees, 2005*; *Shrivastava et al., 2020*). Successful locomotion depends on the execution of body-shape-changes which generate appropriate reaction forces from the terrain, a relationship that can be further complicated if the body motion permanently changes the state of the substrate. Much of our knowledge of terrestrial locomotion is in the regime of rigid materials where the terrain is not affected by passage of the animal (*Koditschek and Full, 1999*; *Sponberg and Full, 2008*; *Kelly et al., 1997*; *Farley et al., 1993*), an understanding which led to the development of robots which are effective on hard ground (*Saranli et al., 2001*; *Liljebäck et al., 2012*; *Blickhan et al., 2007*).

Little is known about locomotion in non-rigid materials which are plastically deformed by the movement of animals or robots, leaving tracks or footprints. Deformable terrains represent a spectrum from materials like water which continuously flow toward the undisturbed, zero shear state (*Lautrup, 2011*) to those which are remodeled by the interaction, like sand. Previous work has mainly focused on motion in fluids, where disturbances dissipate (*Ijspeert, 2008*; *Sfakiotakis et al.,*

*1999*; *Liao, 2007*); the impact of soft material hysteresis on locomotion is not well-understood (*Mazouchova et al., 2013*; *Zhang and Goldman, 2014*).

At one end of the spectrum, where deformations are short-lived (*Boyle, 2010*), small fluid swimmers like the nematode *Caenorhabditis elegans* (*Wen et al., 2012*), spermatazoa (*Gray, 1953*), and bacteria in water (*Rodenborn et al., 2013*) are propelled by the viscous force of the material resisting the animal's body shape changes. In these low-Reynolds-number systems (Re, the ratio of inertial to viscous forces), the inertia of the microscopic animal is negligible compared to the fluid viscosity such that if the animal stops self-deforming it very rapidly stops translating and/or rotating. Such resistive-force-dominated swimming is also found at the macroscale in frictional-fluid swimmers like the sandfish lizard *Scincus scinus* and the shovel-nosed snake *Chionactis occipitalis* moving sub-surface through granular matter (GM) (*Maladen et al., 2009*; *Sharpe et al., 2015*). The motion of these animals is similarly dominated by resistive forces, in this case the frictional interactions between grains. Notably, in both viscous and frictional fluids, the material surrounding the sub-merged swimmers continuously re-flows around the body of the animal such that the animal is always surrounded by material.

At the other end of the spectrum, animals move in materials whose state depends on the history of interaction. Swimmers in fluids at high-Re, where inertia dominates fluid viscosity, can experience time-dependent flow (*Smits, 2019*; *Oza et al., 2019*). Less well-understood is how animals manage material memory (*Keim et al., 2019*) when traversing terrestrial substrates like soil, mud, or sand. Such materials will yield in response to forces applied by a locomotor, similar to a fluid. However, unlike fluids, soft materials can typically bear internal stress. At the free surface, gravity is often insuf-ficient to return mounds of material created by a locomotor to the undisturbed state, exemplified by the tracks left behind by a passing animal. Our previous work (*Mazouchova et al., 2013*; *McInroe et al., 2016*) revealed that performance of limbed animals and robots is affected by the ability to avoid interactions with previously-disturbed material. Using *limbless* lateral undulation on deformable substrates to generate appropriate propulsive forces is a non-trivial task, as demon-strated by the failure of both a snake-like robot (*Maladen et al., 2011a*) and a number of snake spe-cies (*Marvi et al., 2014*) attempting to traverse the surface of GM. However, it is unclear how such animals (or robot models) might manage environmental interactions to move effectively when, unlike limbed locomotors, they cannot increase step size to avoid their own tracks.

In this paper we gain understanding of movement through complex terrestrial materials by com-bining several approaches, each of which provided insight into the number of interrelated elements governing the system. We begin with a study of a variety of non-sand-specialist snake species mov-ing on the surface of GM as well as a desert-specialist snake. We quantify the kinematics of the snakes, which move with varying degrees of success using a range of waveforms. To understand the connection between waveform and performance, as mediated by the GM, we measure the granular response to surface drag. We develop a model for speed dependence that indicates that the granu-lar physics is unchanged over the range of speeds observed in the animal. We then find a character-istic drag anisotropy curve that is largely independent of drag depth and speed. Based on the granular drag experiments and our animal observations we hypothesize that, despite the fast move-ment and complex granular flows, body inertia is negligible. Thus, we introduce a surface granular resistive force theory (RFT). Using RFT, we identify a trade off between actuation speed and torque from the GM that explains the stereotyped waveform used by the sand-specialist as that which maximizes movement speed under an anatomical power constraint. We show that performance of the non-sand-specialist snakes depends on both morphology and waveform, although RFT only accurately predicts performance when lateral slipping of the body is less than a threshold value. In these cases we often observe the animals re-interacting with material remodeled in previous undula-tion cycles. We study the interplay of waveform and material remodeling using a robophysical model. RFT calculation of robot locomotion, as in the high-slip snakes, is inaccurate. Particle image velocimetry (PIV) measurements indicate RFT over-estimates performance when the robot body is pushing material lateral to the direction of motion. RFT does, however, provide insight into the regime of locomotor failure, where we find the desert-specialist operates far from failure due to a combination of its waveform, slender body, low-friction scales, and lifting of segments that only pro-duce drag. We close with a summary of our results and a brief discussion of the implications.

Better understanding of the connection between body self-deformations, substrate remodeling and force generation, and the resulting locomotor performance will elucidate principles for effective

motion on hysteretic materials. This can be applied to the next generation of all-terrain robots. Further, future studies can leverage such knowledge of the benefits and limitations of different body–terrain interaction modes to tease apart neuromechanical control strategies for contending with natural terrains (*Schiebel et al., 2019*).

## Results and discussion

### Performance of limbless locomotors on granular material

Snakes occupy a variety of habitats and display a wide range of morphologies. We took advantage of this natural diversity to explore how different body shapes and patterns of self-deformation fare on GM (granular matter, see *Table 1* for a list of symbols and abbreviations). We studied 22 species of snakes representing five families from the collection at Zoo Atlanta. Snake body plans ranged from short and stout to long and slender (*Figure 1(a,b)*; *Table 2*), and their natural habitats encompassed a broad range from wetlands and swamps, to wet and dry forests and rainforest canopies, to rocky deserts and mountains.

As a counterpoint to the variety of snakes, which were either terrain generalists or specialized to habitats which did not have an omnipresent granular substrate, we also studied the shovel-nosed snake *Chionactis occipitalis* (*Figure 1(c)*; *Table 3*). This species is specialized to bury within (*Kavanau and Kavanau, 1966*; *Sharpe et al., 2015*) and move across (*Mosauer, 1933*) the dry sand of their desert habitat. We used nine individuals collected from the desert near Yuma, Arizona, USA (see Materials and methods for details). *C. occipitalis* uses a stereotyped waveform (*Schiebel et al., 2019*) to quickly move across the sand with little slipping of the body (*Mosauer, 1933*).

**Table 1.** List of frequently-used symbols and abbreviations

| Symbol | Definition |
| --- | --- |
| GM | Granular Material |
| RFT | Resistive Force Theory |
| Re | Reynolds number. Ratio of inertial to viscous forces in a system. |
| $\theta$ | angle between body segment tangent and average direction of motion |
| $\theta_m$ | average maximum $\theta$, attack angle |
| $\xi$ | spatial frequency, number of waves on the body |
| $v_{CoM}$ | center-of-mass velocity |
| $v_{seg}$ | segment velocity (as if riding on the trunk) |
| $\omega_{CoM}$ | angular velocity about the center-of-mass |
| $\beta_d$ | granular drag angle |
| $v_d$ | plate drag velocity |
| $z$ | depth intrusion into GM, measured from the free surface |
| $\rho$ | density of granular material |
| $\beta_s$ | snake body slip angle |
| $\sigma_n$ | component of granular stress normal to area element |
| $\sigma_t$ | component of granular stress tangential to area element |
| $\sigma_n/\sigma_t$ | anisotropy factor. Ratio of normal to tangential stress |
| $\mu$ | scale-GM friction coefficient |
| $L$ | total body length |
| $L/w$ | Aspect Ratio, snake body length divided by width at the widest point |
| $g$ | gravitational constant = 9.81 ms$^{-2}$ |
| $s$ | arclength along snake body midline measured from the head |

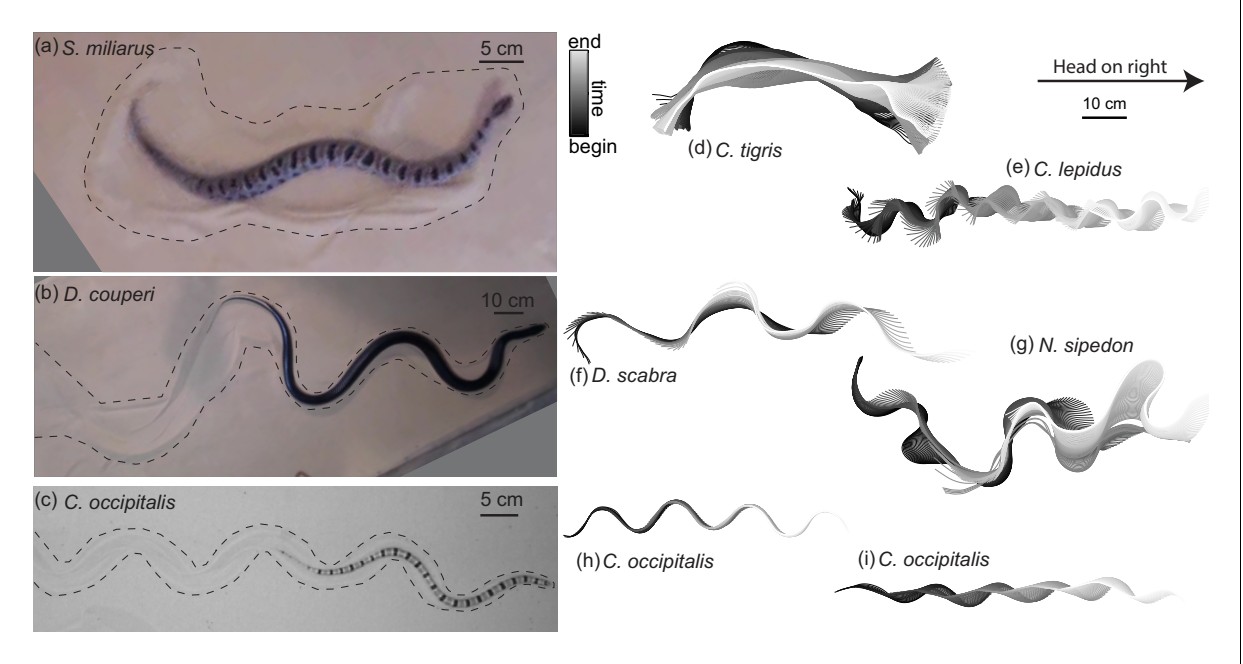

**Figure 1.** Body shape, waveform, and ability to progress across GM varies among snake species. (**a–c**) Snapshots of snakes moving on the surface of GM. Dashed lines roughly indicate the area of material disturbed by the motion of the animal. (**a**) The generalist pygmy rattlesnake *Sistrurus miliarius* attempting to move on natural sand collected from Yuma, Arizona, USA. The animal has completed several undulations, sweeping GM lateral to the midline of the body. This snake failed to progress further than pictured. (**b**) The generalist eastern indigo snake *Drymarchon couperi*. On the same GM as (**a**) (**c**) The sand-specialist shovel-nosed snake *Chionactis occipitalis* in the lab on 300 μm glass particles. (**d–i**) Digitized midlines of animals. Color indicates time from beginning to end of the trial. All scaled to the 10 cm scale bar shown. (**d–g**) are on Yuma sand and (**h,i**) are on glass particles. (**d**) Tiger rattlesnake *Crotalus tigris*, a rocky habitat specialist. The animal was unable to progress on the GM. Total length of the trial $t_{tot}$ = 25.7 s, time between plotted midlines $\delta t$ = 100 ms (**e**) Rock rattlesnake *Crotalus lepidus*, a rocky habitat specialist. $t_{tot}$ = 9.4 s, $\delta t$ = 33 ms (**f**) Egg-eating snake *Dasypeltis scabra* inhabits a wide range of habitats in Africa. $t_{tot}$ = 1.6 s, $\delta t$ = 33 ms (**g**) *Nerodia sipedon*, a water snake inhabiting most of the Eastern US extending into Canada. $t_{tot}$ = 6.0 s, $\delta t$ = 33 ms (**h**) *C. occipitalis* 130 and (**i**) 128. These trials represent the inter-individual variation in kinematics. Some of the animals moved approximately 'in a tube' where all segments of the body followed in the path of their rostral neighbors (e.g. (**h**), $t_{tot}$ = 1.25 s, $\delta t$ = 12 ms) while others used this strategy only on the anterior portion of the body, appearing to drag the posterior segments in a more or less straight line behind themselves (e.g. (**i**), $t_{tot}$ = 1.01 s, $\delta t$ = 12 ms).

## Kinematics and ability vary among species

The non-desert specialist snakes from the collection at Zoo Atlanta were tested in a 2 × 1 m² air-fluidized bed filled with sand collected in Yuma County, Arizona, USA. The snakes' movement was captured at 30 frames per second (fps) using an overhead camera (data collected for *Marvi et al., 2014*). The desert-specialists were tested in an air-fluidized trackway of area 152 × 53 cm² filled with 297 ± 40 μm particles (Potters Industries spherical glass Ballotini beads). *C. occipitalis* moved quickly relative to the non-sand-specialists so its kinematics were captured at 250 fps. In all experiments, a blower attached to the bottom cavity of the fluidized bed was turned on and air flow increased until the GM was in a fluid-like state. The air flow was then decreased until the GM settled in a loose packed state (as described in *Maladen et al., 2009*). Air flow was always off during experiments and the penetration depth of the snakes (O(mm)) was much less than the total depth of the GM (O(cm)).

We used custom MATLAB code to digitize the midlines of the snakes; image processing functions identified the non-sand-specialist and the method described in *Sharpe et al., 2015* tracked the sand-specialist's banded black markings (*Schiebel et al., 2020*). A cubic spline fit to the tracked data evenly divided the snake midlines into 100 measurements over the entire body (non-sand-specialists) or from neck to vent (*C. occipitalis*).

The use of alternating left and right bends, typical of lateral undulation in snakes, on the body was ubiquitous, although specifics of the waveform such as amplitude and number of bends on the body varied (*Figure 1*). Among the non-specialists, some snakes used uniform, periodically repeating

**Table 2.** Anatomical information for the non-sand-specialist species.

| Species | Subfamily | Family | Length (cm) | Max width (cm) | Mass (g) |
|---|---|---|---|---|---|
| *Acranthophis dumerili* | | Boidae | 183 | 7.9 | 5620 |
| *Agkistrodon bilineatus* | Crotalinae | Viperidae | 75.4 | 3.3 | 306.7 |
| *Agkistrodon contortrix* | Crotalinae | Viperidae | 80 | 3.1 | 359.5 |
| *Agkistrodon piscivorus* | Crotalinae | Viperidae | 92 | 4.5 | 569.5 |
| *Agkistrodon piscivorus* | Crotalinae | Viperidae | | | 874.5 |
| *Agkistrodon piscivorus* | Crotalinae | Viperidae | 57 | 2.7 | 161 |
| *Aspidites ramsayi* | | Pythonidae | 1.3 | 3.4 | 749 |
| *Bothriechis schlegelii* | Crotalinae | Viperidae | 66.7 | 1.7 | 104 |
| *Crotalus lepidus* | Crotalinae | Viperidae | 43 | 2.2 | 44 |
| *Crotalus molossus* | Crotalinae | Viperidae | 82.1 | 4 | 370.7 |
| *Crotalus tigris* | Crotalinae | Viperidae | 71.1 | 3.1 | 331.6 |
| *Crotalus willardi* | Crotalinae | Viperidae | 47 | 3 | 134.5 |
| *Dasypeltis scabra* | | Colubridae | 71 | 1.1 | 51 |
| *Drymarchon couperi* | | Colubridae | 162 | 4.2 | 1018 |
| *Epicrates subflavus* | | Boidae | 153 | 3 | 738 |
| *Lampropeltis getula* | Colubrinae | Colubridae | 121 | 3 | 680 |
| *Lichanura trivirgata* | | Boidae | 71 | 2.6 | 243.5 |
| *Loxocemus bicolor* | | Locoxemidae | 111 | 3.3 | 607.5 |
| *Nerodia sipedon* | Natricinae | Colubridae | 79 | 3.5 | 453.5 |
| *Senticolus triaspis* | Colubrinae | Colubridae | 101 | 1.8 | 198 |
| *Sistrurus catenatus* | Crotalinae | Viperidae | 52 | 2.8 | 174.5 |
| *Sistrurus miliarius* | Crotalinae | Viperidae | 47 | 2.8 | 146 |

bends (e.g. *Figure 1 (b,e,f)*) like *C. occipitalis* (*Figure 1 (c,h,i)*). Others used bends of varying amplitude and wavelength along the body (*Figure 1(g)*).

We used the tangent angle, $\theta$, the angle between the local body tangent vector and the average direction of motion of the animal (*Figure 2* (a,*left*)), to represent body posture at each instant in time. Consistent with previous work (*Schiebel et al., 2019*), we found *C. occipitalis* kinematics were well-described using a serpenoid curve (*Figure 2—figure supplement 1*; *Hirose, 1993*),

**Table 3.** Anatomical information for the individual sand-specialist snakes.

The species *Chionactis occipitalis* is in the family Colubridae. Average and standard dev. of lengths 38.0±1.3 cm.

| Individual | Length (cm) | Max width (cm) | Mass (g) |
|---|---|---|---|
| 120 | 36.4 | 1.1 | 20 |
| 122 | 39.2 | 1.0 | 21 |
| 123 | 40.1 | 1.1 | 20 |
| 124 | 38.1 | 1.1 | 20 |
| 125 | 36.6 | 1.0 | 18 |
| 128 | 38 | 1.0 | 16 |
| 129 | 39.3 | 1.2 | 24 |
| 130 | 37.1 | 1.0 | 20 |
| 132 | 37.1 | 1.0 | 18 |

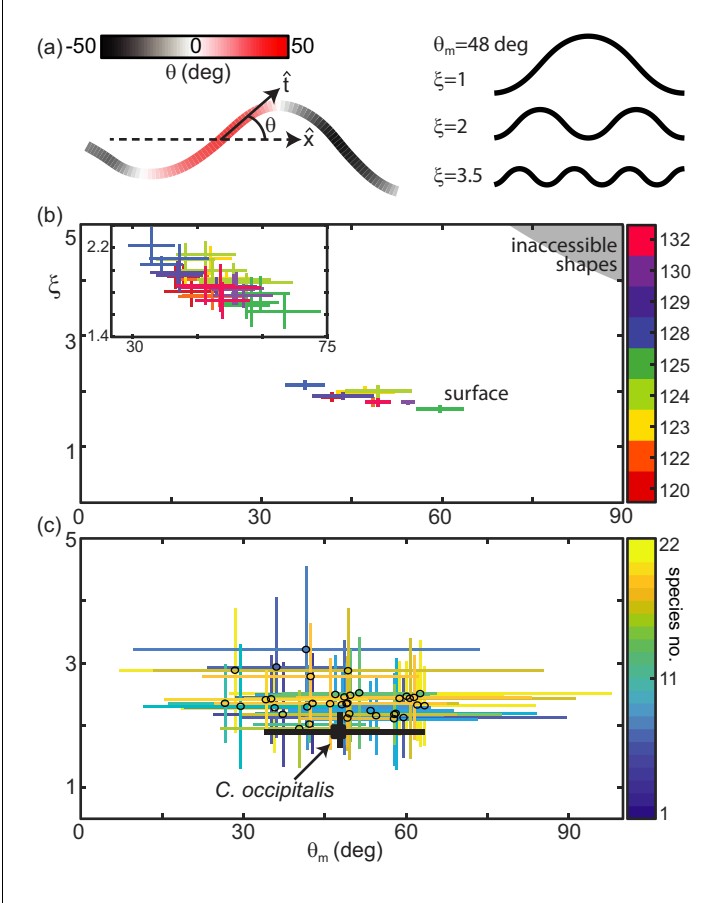

**Figure 2.** Snake waveform parameters measured in experiment. (**a**, *left*) $\theta = \mathrm{acos}(\hat{x} \cdot \hat{t})$ for local tangent angle, $\hat{t}$ and average direction of motion $\hat{x}$. Example body posture shown, colored by $\theta$. (**a**, *right*) Example serpenoid curves (*Equation 1*) of different $\xi$ on a body of fixed length and $\theta_m = 48$ deg. (**b**) *C. occipitalis* experimental measurements plotted in the $(\theta_m, \xi)$ parameter space. Color indicates animal number. Markers are the mean and range of each individual taken over all trials. N = 9 individuals, n = 30 trials. The gray region in the upper right corner are waves which are inaccessible given the flexibility of the snake (*Sharpe et al., 2015*). $\theta_m$ was comparable to values measured from images of tracks taken in the field (*Figure 2—figure supplement 2*). (*inset*) A close-up of the data with mean and range from each trial plotted individually. Color consistent with the main plot (Linear fit with 95% confidence interval to inset data: slope −0.016 (-0.020,–0.011) intercept 2.66 (2.42, 2.90), R-square = 0.6). (**c**) Axes as in (**b**), black cross is the combined *C. occipitalis* measurements. Colored crosses are the mean and range of values measured on a per-trial basis in the non-sand-specialists, different species indicated by color. N = 22 individuals, n = 38 trials.

The online version of this article includes the following figure supplement(s) for figure 2:

**Figure supplement 1.** Principal component analysis (PCA) of all sand snake shapes.
**Figure supplement 2.** Kinematics of *C. occipitalis* compared to field measurements.

$$\theta(s,t) = \theta_m \sin\left(\frac{2\pi\xi}{L}(s + v_{seg}t)\right). \tag{1}$$

The speed that the wave propagates down the body, $v_{seg}$, determines how quickly the shape changes in time, $t$. At any moment, the shape of the snake as a function of arclength along the body, $s$, is determined by the amplitude, $\theta_m$, also referred to as the attack angle, and spatial frequency, or number of complete waves on the body, $\xi$ (*Figure 2(a)*, right). For an infinitesimally thin curve with infinite resolution these parameters are independent. In practice, the musculoskeletal system and width of the body limits both how quickly the midline can change curvature and how sharply bent the body can be, although sufficiently far away from these limits $\theta_m$ and $\xi$ remain independent. While

there may be other, not purely anatomical, factors which connect these variables, in this work we chose to measure their values and study the connection to performance as mediated by the GM. Future study could explore the neuromechanical origin of the shapes and its connection to the differences observed between different species.

Inspired by this two-parameter description, we characterized snake waveforms using $\theta_m$ and $\xi$. We calculated attack angle and wavenumber at each frame in a trial (see Materials and methods for details) and took the mean over all frames to obtain $\theta_m$ and $\xi$ representative of the average snake posture. Plotting the results for each of the nine *C. occipitalis* in the $\xi$ versus $\theta_m$ parameter space revealed that the animals used a limited subset of the shapes they were anatomically capable of adopting (*Figure 2(b)*). There was a slight downward trend in the relationship between $\xi$ and $\theta_m$ when considering both each individual and each trial (*Figure 2(b)*, inset). Given the weak relationship between $\xi$ and $\theta_m$, we combined all individual measurements and examined the impact of the average *C. occipitalis* wave, taken as the mean and range of the averages shown (*Figure 2(c)*, black cross).

Compared to *C. occipitalis*, and consistent with observation (*Figure 1*), there was greater variety among the wave parameters measured on the non-sand-specialist species (*Figure 2(c)*). However, as in *C. occipitalis*, the snakes did not fill the space of anatomically possible shapes. The various combinations of waveforms and performance suggested the forces acting on the snakes depended, at least in part, on the shape used, while the limited space of waveforms alluded to the presence of internal and/or external constraints.

We observed deformations in the granular substrate caused by the motion of the animals (*Figure 1(a–c)*). This indicated that the shape changes carried out by the animals yielded the media, a situation which necessarily resulted in reaction forces acting on the animal from the GM. We thus explored the connection between body segment motion and the consequent forces by carrying out granular drag experiments using a partially submerged plate.

## Granular drag measurements

Principles governing subsurface swimming in GM were elucidated by granular drag measurements (*Maladen et al., 2009*). These experiments revealed that during subsurface swimming in GM the material behaved as a frictional fluid. Like a low-Re swimmer in a viscous fluid, propulsive forces arose from the resistance of the GM to the animal's shape changes (although rather than viscous forces the grains interact with each other via normal and frictional contacts *Maladen et al., 2009*; *Zhang and Goldman, 2014*).

We observed the formation of granular piles as the snakes self-deformed on the granular surface (*Figure 3 (a)*, *Figure 3—video 1*). This suggested that, consistent with subsurface swimming (*Maladen et al., 2009*) and surface walking (*Mazouchova et al., 2013*; *McInroe et al., 2016*), slithering animals propel themselves using the granular forces arising from the body yielding the material (and not solely frictional anisotropy of the ventral scutes *Hu et al., 2009*). The interaction between the body and the GM can be characterized by the amount of slip, $\beta_s = \mathrm{acos}(|\hat{t} \cdot \hat{v}|)$, relative to the substrate, where $\hat{t}$ and $\hat{v}$ are the local velocity and tangent unit vectors of a body segment (*Figure 3 (a)* inset; *Sharpe et al., 2015*). $\beta_s$ emerges from the interrelationship between the granular stress and the animal's self-deformation pattern.

We empirically measured the granular stress on a simple model for a snake body segment—an aluminum plate commanded to move at drag angle $\beta_d$ between the drag velocity unit vector $\hat{v}_d$ and the plate face tangent in the horizontal plane (*Figure 3 (a,b)*). An aluminum rod attached the Al plate to a force transducer (ATI nano43) that decomposed stresses into those acting normal $\sigma_n$, and tangent, $\sigma_t$, to the plate face (*Figure 3 (c)*). This apparatus was located on the end effector of a six degree-of-freedom robot arm (Denso VS087A2-AV6-NNN-NNN) which controlled intruder movement. The robot arm first rotated the plate to $\beta_s$, then submerged it to depth $z$, measured from the free surface to the plate's bottom edge (*Figure 3 (b)*), next it dragged the plate parallel to the surface for 20 cm at constant speed, and lastly stopped and extracted it. A fluidizing bed containing the same 297 ± 40 μm glass particles used in the *C. occipitalis* experiments prepared the material to an initially loose-packed state using two shop vacs controlled by a proportional relay. The fluidization procedure was the same as in the snake trials.

Unlike subsurface drag stresses, which developed almost instantaneously to the steady state (*Maladen et al., 2009*), at the surface stress monotonically increased over several centimeters before

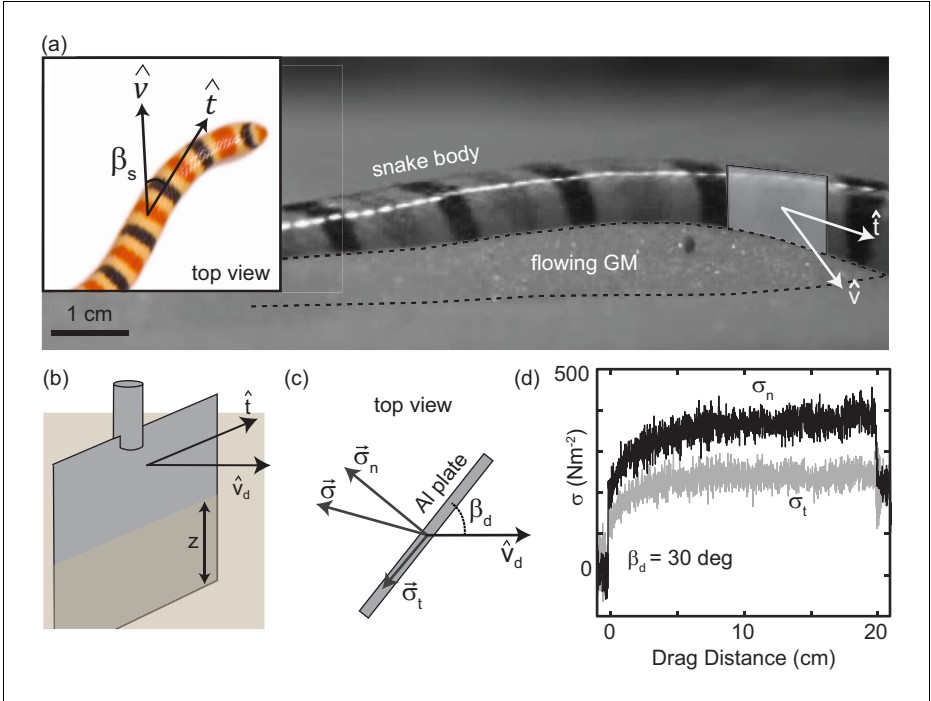

**Figure 3.** Characterizing granular forces experienced by snakes by empirically measuring stress on a partially buried plate. (a) Side-view of *C. occipitalis* moving on the surface. Snake is moving from left to right. The surface was initially featureless, the piles of sand outlined by the black, dashed line were generated by the motion of the snake. Gray rectangle illustrates the flat plate snake body segment model, aligned with $\hat{t}$ of the midline moving in direction $\hat{v}$. (*inset*) Top view of a snake. $\beta_s$ is the angle between the local tangent and velocity unit vectors. (b) Aluminum plate, dimensions $3 \times 1.5 \times 0.3$ cm$^3$. The plate was kept at a constant depth, $z$, from the undisturbed free surface of the GM to the bottom of the intruder. (c) Top view of the plate moving in a direction $\hat{v}_d$ at angle $\beta_d$. Components $\sigma_t$ and $\sigma_n$ of the total stress, $\sigma$. (d) Raw drag data collected at $\beta_d = 30$ deg and $v_d = 10$ mm s$^{-1}$ as a function of drag distance of the plate. The upper black curve is $\sigma_n$, the lower gray is $\sigma_t$. Force data collected at 1000 Hz, plotted here down-sampled by a factor of 10 to facilitate rendering.

The online version of this article includes the following video for figure 3:

**Figure 3—video 1.** Side view of *C. occipitalis* moving on GM.
https://elifesciences.org/articles/51412#fig3video1

saturating (*Figure 3(d)*). This is due to the free surface flow of the GM; a pile of sand above the surface, like those created by the snakes, appeared at the leading face of the intruder at the onset of drag and increased in volume until reaching a balance between the new grains being encountered and those flowing around the edges of the plate. Previous studies of plate drag at the surface (at $\beta_d = 90$ deg) measured a similar drag-distance dependent force and observed a wedge-shaped region of GM, beginning at the bottom edge of the plate and extending above the surface, whose constituent grains were flowing forward and up against gravity (*Gravish et al., 2014*; *Guo et al., 2012*).

## Velocity-dependent stress captured by grain inertia model

The snake speeds were variable ($v_{seg}$ from $\approx 35$ to 95 cms$^{-1}$, *Figure 4—figure supplement 1*) and the intrusion depth of the snakes' trunk into the GM ranged from 0 (no intrusion, occurring at the apexes of the wave which the snake lifted off of the surface) to $\approx 5$ mm (*Figure 4—figure supplement 2*). Previous studies indicated that granular drag stress depends on both intruder speed (*Percier et al., 2011*) and depth (*Albert et al., 1999*; *Marvi et al., 2014*).

Commensurate with those studies, we fixed $\beta_d = 90$ deg and $z = 8$ mm and found normal stress quadratically increased as $v_d$ increased from 1 mm s$^{-1}$ to 750 mm s$^{-1}$ (the limit of the robot arm capability, *Figure 4(a)*). Similarly, for $\beta_d = 90$ deg and $v_d = 10$ mm s$^{-1}$, normal stress linearly increased

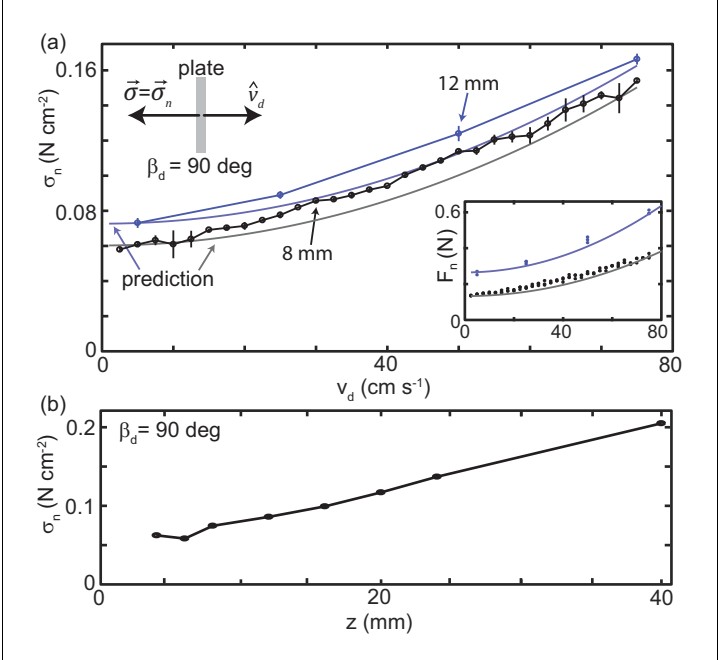

**Figure 4.** Granular drag stress as a function of speed and depth. (a) Stress normal to the plate face as a function of $v_d$ at a constant depth $z = 8$ mm (black points) and $z = 12$ mm (blue points). As shown in the diagram at the upper left, $\beta_d = 90$ deg for all trials such that the total stress was equal to $\sigma_n$. Markers are mean and std. of three trials. Where error bar is not visible error is smaller than the marker. Gray curve is the model prediction for $z = 8$ mm and light blue curve for $z = 12$ mm. (inset) Normal force versus $v_d$ measured in experiment (circle markers) and as predicted by the model (solid curves). Color is consistent with the main plot. Each dot is the average force measured in one trial, all trials shown. (b) Stress normal to the plate face as a function of $z$ at $v_d = 10$ mm s$^{-1}$. All experimental data shown in this figure are averaged from 10 to 20 cm drag distance. Over this distance the force had reached steady state and the robot arm was moving at the commanded velocity.

The online version of this article includes the following figure supplement(s) for figure 4:

**Figure supplement 1.** *C. occipitalis* movement speed independent of waveform parameters.
**Figure supplement 2.** *C. occipitalis* tracks.

---

with $z$ from 4 mm, the shallowest depth where force could be resolved, to 40 mm, where the plate was fully submerged with the top edge 10 mm below the surface (*Figure 4(b)*).

For snakes moving slowly on non-deformable surfaces, body inertia was small compared to the frictional forces between the belly and the surface (*Hu et al., 2009*). While *C. occipitalis* was moving quickly, we observed, in line with earlier reports (*Gray, 1946*), that forward motion would immediately cease when the animals stopped propagating the wave. This phenomenon is observed in swimmers at low-Re where the dominant resistive forces of the surroundings inhibit gliding.

The animal behavior indicated that friction between the body and the GM and local, dissipative interactions within the GM were the dominant forces in the system. Additionally, once the drag experiment was in the steady state, we did not observe any time-dependent structures in the GM. Thus, we used the ansatz that stress on the plate could be modelled as independent, time-invariant static stress and material inertia terms,

$$\sigma = \sigma_o + \rho v_d^2. \tag{2}$$

The $\rho v_d^2$ term in the formulation is the momentum transfer requirement for pushing the granular material ahead of the plate element, and the static ($v_d \to 0$) stress term, $\sigma_o$, is the stress required to initiate yielding of the media. $\rho$ is the density of the material, which we estimated using the density of glass, $\rho_{glass}$ and the packing fraction $\phi = 0.64$ induced by motion of the intruder, $\rho = \phi\rho_{glass} = 0.64 \times 2500 \text{ kgm}^{-3}$.

The static stress on the intruder may be modeled using gravitational loading of the grains. In fluids, hydrostatic pressure is given by $\rho g z_l$ given gravity $g$ and local plate element depth $z_l$. In granular materials there is a similar depth dependence, however, because the bulk can harbor internal stresses, the behavior differs substantially and $\sigma_o = B\rho g z_l$ where $B$ can be an order of magnitude greater than in a fluid (**Brzinski et al., 2013**). In our experiment, the plate extends from the free surface to depth $z$, so $z_l = 0.5z$. We assume $B$ is $O(10)$ to estimate that at $z = 8$ mm ($z_l = 4$ mm) $\sigma_o \approx 10\rho g z_l = 0.06$ Ncm$^{-2}$ and at $z = 12$ mm $\sigma_o \approx 0.09$ Ncm$^{-2}$ which are similar to the experimentally measured values.

$B$ can be calculated more precisely using, for example, Coulomb-wedge theory (**Guo et al., 2012**) or plasticity theory (**Kang et al., 2018**). However, for the purposes of this study, we were interested in understanding the stress dependence on drag speed and the implications for animal locomotion. Thus, to facilitate comparison with the momentum transfer term, the curves plotted in **Figure 4** were calculated using $\sigma_o$ estimated from the experimental data. $\sigma_o$ was calculated by subtracting $\rho v_d^2$ from $\sigma_n$ measured at the three lowest speeds collected at a given $z$ and averaging the result.

Using the empirically obtained static stress, the $\rho v_d^2$ term captured stress as a function of $v_d$ without any free parameters (**Figure 4(a)**). This was consistent with results for subsurface drag (**Brzinski III and Durian, 2010**), indicating that, despite the free surface, the physics governing material stresses was similar. While the bulk of the effect of speed was captured, the model was not exact. Future study could explore the origin of this deviation, however, a more thorough investigation of the granular physics is beyond the scope of this paper. Regardless, the predictive power of this simple, time-independent model supported our hypothesis that, across the relevant drag speeds, stress was the result of localized interaction with the material.

Eels in water (a high-Re system) increase the CoM velocity, $v_{CoM}$, by both increasing the speed of wave propagation and altering the waveform to manage the speed-dependent fluid response (**Tytell, 2004**). *C. occipitalis* moving on the surface had a linear relationship between the speed of wave propagation and $v_{CoM}$ (**Figure 4—figure supplement 1(a)**) while the waveform was independent of $v_{CoM}$ (**Figure 4—figure supplement 1 (b,c)**). This suggested that, despite the $v_d^2$ stress dependence, the aggregate forces responsible for locomotion were not a function of wavespeed but depended only on the pattern of self-deformation. We thus explored the ratio of thrust to drag forces as a function of plate orientation, depth, and speed.

## Drag anisotropy is not strongly dependent on speed or depth

At low-Re, the normal and tangential stresses acting on segments of long, slender swimmers in Newtonian fluids are approximated by $\sigma_n = C_n \sin(\beta_d)$ and $\sigma_t = C_t \cos(\beta_d)$, respectively (**Gray, 1955**). The constants $C_n$ and $C_t$ are determined by the geometry of a body segment and the viscosity of the surrounding fluid, and the ratio $C_n/C_t$ can be used to approximate swimming performance of a given waveform (**Boyle, 2010**). Subsurface sand-swimming performance is similarly determined by the balance of thrust to drag, although the prefactors are functions of $\beta_d$ (**Maladen et al., 2009**; **Maladen et al., 2011b**). Given our hypothesis that the surface slithering snakes were resistive-force-dominated like a low-Re swimmer, we measured the ratio of thrust to drag stress, $\sigma_n/\sigma_t$. Based on our animal observations and results from the speed-dependent drag experiments, we expected that this ratio depended on drag angle but not drag speed or depth.

We measured stress as a function of $\beta_d$ ranging from 0 deg to 90 deg (**Figure 3(b)**). The ratio $\sigma_n/\sigma_t$ was largely independent of the drag distance (**Figure 5(a)**), especially at small $\beta_d$. There was a periodic fluctuation of $<1$ $\sigma_n/\sigma_t$ occurring over several cm appearing in all trials. However, both these fluctuations and the slope of $\sigma_n/\sigma_t$ as a function of drag distance were small relative to the effect of changing $\beta_d$. Thus, we averaged $\sigma_n$ and $\sigma_t$ to characterize the relationship between plate orientation and stress normal and tangential to its face (**Figure 5(b)**).

Normal stress increased monotonically with $\beta_d$ while tangential stress was approximately constant, gradually falling to zero as $\beta_d$ went to 90 deg (**Figure 5(b)**). The ratio $\sigma_n/\sigma_t$ as a function of $\beta_d$ for both varying speeds and depths collapsed to a characteristic anisotropy curve (**Figure 6(a)**). The same drag anisotropy curves were measured in poppy seeds and oolite sand and were similarly independent of depth and speed (**McInroe et al., 2016**). The appearance of this curve in diverse GM suggests it may be a more general feature of dissipative, deformable materials.

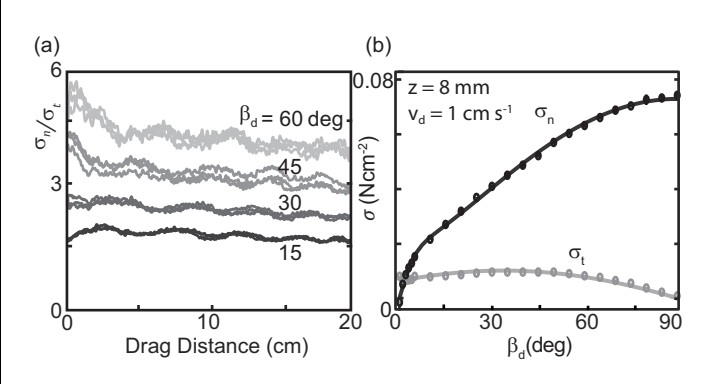

**Figure 5.** Stress as a function of plate drag angle. (a) Stress anisotropy versus drag distance. Since $\sigma_t$ is small, values with noise included can approach zero. As we were interested in the force evolution over several cm (for the plotted drag speed this corresponds to signals <1 Hz), we removed fluctuations above 5 Hz using a low-pass butterworth filter. Curves from three trials are shown for each of four different, fixed $\beta_d = 15$ (darkest, bottom curve), 30, 45, and 60 deg (lightest, top curve) as labeled on the right of the plot. The slopes of a linear regression fit to the average of three trials as a function of drag distance were $-1.8 \times 10^{-5}$, $-2.9 \times 10^{-5}$, $-5.0 \times 10^{-5}$ and $-6.4 \times 10^{-5}$ cm$^{-1}$ for $\beta_d = 15$, 30, 45, and 60 deg, respectively (R-square = 0.6, 0.8, 0.8, and 0.7). (b) Average stress as a function of changing drag angle. Each measurement was calculated by averaging the raw stress data from 10 to 20 cm drag distance. Each data point is the mean and standard deviation of three trials; error bars are smaller than the markers. Solid lines are the fit functions used in the RFT calculations.

In viscous fluids the constants $C_n$ and $C_t$ are independent of drag angle such that $\frac{\sigma_n/\sigma_t}{\tan(\beta_d)}$ is constant. Consistent with results for subsurface drag (*Maladen et al., 2009*; *Maladen et al., 2011b*), we find that at the surface $\frac{\sigma_n/\sigma_t}{\tan(\beta_d)}$ is a nonlinear function of $\beta_d$ (*Figure 6(b)*). Because tangential stress magnitudes were relatively small and, unlike in viscous fluid, normal stresses rose sharply with $\beta_d$ at small angles (*Figure 5(b)*), thrust and drag forces on the plate were equal ($\sigma_n/\sigma_t = 1$) at smaller $\beta_d$ than in viscous fluid or subsurface in GM (*Figure 6(a)*). Reflecting the small angles where the anisotropy was unity, $\beta_s$ measured on the snake were small (*Figure 6(a)*, gray histogram).

Plotting anisotropy at constant $\beta_d$ as speed and depth varied further illustrated the relatively weak dependence of $\sigma_n/\sigma_t$ on these variables (*Figure 6(c,d)*). Especially at small $\beta_d$, anisotropy did not depend on $v_d$ (*Figure 6(c)*). $\sigma_n/\sigma_t$ was more dependent on $z$ at shallow depths, decreasing twofold as $z$ increased from 4 to 8 mm (*Figure 6(d)*).

The invariance of $\sigma_n/\sigma_t$ with speed was in accord with our hypothesis of locomotor performance that was dependent solely on the pattern of self-deformation. The efficacy of the simple $\rho v_d^2$ model of the relationship between speed and granular stress indicated that time-dependent structures like vortices were not present, supported by observation of the fast dissipation of disturbance to the grains in both animal and drag experiments. These results indicated that resistive force theory (RFT, *Gray, 1955*), in particular the same speed-independent granular RFT used to predict subsurface performance in GM (*Maladen et al., 2009*; *Zhang and Goldman, 2014*), was a viable candidate for modelling snake motion on the granular surface.

## Surface resistive force theory model

Continuum constitutive equations for granular media (similar to Navier-Stokes equations for fluids) can predict granular flows (*Kamrin and Koval, 2012*). However, these methods, as well as established discrete element methods (like MD), are computationally expensive. Granular resistive force theory, which relies on the assumption that the total force on the body is a linear, independent superposition of the forces on individual segments (*Gray, 1955*), is by comparison computationally cheap. RFT successfully modeled a number of systems for which these assumptions are valid, although its effectiveness in a system exhibiting hysteresis was unknown (*Zhang and Goldman, 2014*).

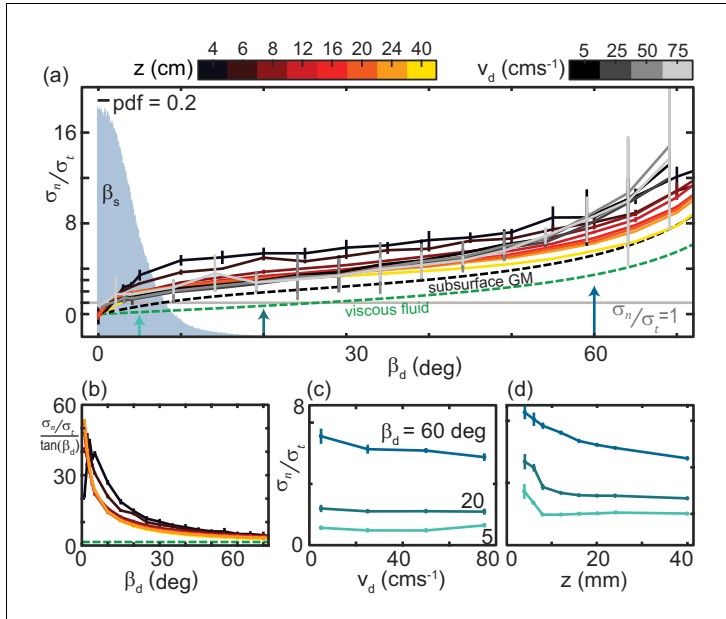

**Figure 6.** The ratio of granular normal to tangential stress, $\sigma_n/\sigma_t$, is not strongly dependent on speed or depth. (a) $\sigma_n/\sigma_t$ as a function of $\beta_d$. Light gray horizontal line indicates $\sigma_n/\sigma_t = 1$. All depths and speeds are plotted as indicated by color. Dashed curves are the anistropy for a low-Re swimmer in Newtonian fluid (green, $C_n/C_t \approx 1.5$ for a smooth, long, and slender swimmer **Boyle, 2010**) and subsurface in the same $\approx 300$ μm GM used in this study (black). Labels are below the associated curves. The solid gray area is the probability density (pdf) of $\beta_s$ measured on the snake in experiment (N = 9 individuals, n = 30 trials). (b) $\frac{\sigma_n/\sigma_t}{\tan(\beta_d)}$ versus $\beta_d$ for the surface granular drag at varying $z$ and viscous fluid shown in (a). Line color and types consistent with (a). (c) $\sigma_n/\sigma_t$ as a function of $v_d$ at the values of $\beta_s$ indicated. Colors correspond to the colored arrows on the horizontal axis in (a). Linear fits to the data (not shown) with slope, m, and 95% confidence bounds in parentheses for $\beta_d = 60$ deg $m = -0.019$ deg $(-0.040, 0.003)$ R-square = 0.88; $\beta_d = 20$ deg $m = -0.003$ deg $(-0.007, 0.002)$ R-square = 0.75; $\beta_d = 5$ deg $m = 0.003$ deg $(-0.011, 0.016)$ R-square = 0.23, (d) $\sigma_n/\sigma_t$ as a function of $z$. Colors and angles are the same as in (c).

We observed that as a trunk segment pushed laterally against the sand it built a pile of grains, like that which evolved during drag of the plate, on the side of the body closest to the tail (**Figure 3—video 1**). The side of the body opposite the pile (closest to the head) appeared to be in contact with little material. Therefore, we chose to model the snake body as a flat plate, representing the caudal-facing side of the body, with an added term to account for drag on the ventral surface. This is not an unreasonable model for this species, as the ventral surface is known to be flat or even slightly concave (**Mosauer, 1933**). As the snake held the head slightly raised from the surface we did not include a term accounting for head drag.

We approximated the kinematics of the snake using $n_{segs} = 100$ segments evenly distributed along the arclength ($s$) at 70 points in time ($t$) divided evenly over one full undulation cycle. At this resolution adding or subtracting five segments or time steps changed the velocity prediction by less than 0.1%. We calculated each segment's orientation and velocity given parameters $\theta_m$ and $\xi$ using the equation for a serpenoid curve (**Equation 1**). The segment speed, $v_{seg}$, was 1 ms$^{-1}$ unless otherwise stated.

The force on a segment of length $ds = \frac{L}{n_{segs}}$ was assumed to depend only on its movement through the material, $\beta_d$, given by its orientation ($\hat{n}$, $\hat{t}$) and direction of motion, ($\hat{v}$). Velocity, $\vec{v}(\vec{v}_{seg}, \vec{v}_{CoM}, \vec{\omega}_{CoM})$, was determined by the local segment velocity, $\vec{v}_{seg} = v_{seg}\hat{t}$, added to the center-of-mass velocity, $\vec{v}_{CoM}$, and rotation about the CoM, $\vec{\omega}_{CoM}$. $\hat{n}$ and $\hat{t}$ were calculated using **Equation 1**. Given functions $f_n(\hat{n} \cdot \hat{v})$ and $f_t(\hat{t} \cdot \hat{v})$ relating granular force normal and tangential to the segment, respectively, to the segment's motion, the infinitesimal force was thus

$$dF = \frac{|z_{snake}|ds}{|z|w_{plate}}(f_n(\hat{n} \cdot \hat{v})\hat{n} + f_t(\hat{t} \cdot \hat{v})\hat{t}). \tag{3}$$

$z$ was the plate intrusion depth (as defined in *Figure 3*), $w_{plate}$ was the width of the plate used in the granular drag experiments, and $z_{snake}$ was the snake intrusion depth. Unless otherwise noted, we assumed $z_{snake} = z = 8$ mm. This prefactor scales the granular forces under the assumption that force was linearly related to intruder area. While shape effects on drag at the surface are not well known, during subsurface drag the horizontal forces on an object depend primarily on cross-sectional area with little dependence on the shape of the object, much less than in fluids (*Albert et al., 2001*). There are also shape dependent lift forces on an immersed intruder (*Ding et al., 2011*), however, we assumed that vertical forces acting on the snake were balanced and did not affect the horizontal forces determining performance. Our depth-dependent drag experiments, which revealed drag stress increased linearly with depth (*Figure 4*), also indicated force scaled linearly with area.

The function $f_n$ was given by

$$f_n(\hat{n} \cdot \hat{v}) = A(1 + \frac{C}{\sqrt{\tan^2\gamma_o + (\hat{n} \cdot \hat{v})^2}}), \tag{4}$$

where $A, C$, and $\tan^2\gamma_o$ were constants determined by fitting the drag data. For example, at $z = 8$ mm and $v_d = 10$ mms$^{-1}$, (i.e. data in *Figure 5(b)*) $A = 0.15$ N, $C = 0.198$, and $tan^2\gamma_o = 0.00389$. This is similar to the function presented in *Sharpe et al., 2015*, with the addition of the overall scaling constant $A$. We found that $f_t$ was not well-captured by our previous subsurface models. In the interest of carrying out the RFT calculation, we thus used a Fourier fit to the data to determine $f_t$ (*Table 4*).

We modeled ventral drag on each segment as acting opposite the velocity with magnitude $\mu mg/n_{segs}$ where $\mu$ was the coefficient of friction between the snake's ventral scutes (scales on the belly) and the substrate, $m$ was the snake mass, and $g$ was the gravitational constant.

Previous research (*Hu et al., 2009*) found that the ventral scutes were anisotropic such that gliding directly forward produced less drag than sliding laterally, and both forward and lateral motion resulted in less frictional drag than moving directly backwards. The coefficient of static friction between *C. occipitalis* scales and the glass beads was found to be $0.109 \pm 0.016$ when sliding forward and $0.137 \pm 0.018$ backward (*Sharpe et al., 2015*). The lateral coefficient of friction for *C. occipitalis* scales is not known. However, given the results of *Hu et al., 2009*, we assumed that it was bounded between 0.10 and 0.14. Thus, the difference between frictional forces acting tangential and normal to segments was negligible compared to the forces due to the motion of the body through the GM Therefore, we chose not to include the frictional anisotropy in our RFT calculation.

We previously found the coefficient of static friction between aluminum and the glass particles was approximately 0.2 (*Maladen et al., 2009*). To account for the difference between the plate friction and snake-scale friction, we scaled the tangential drag forces by the ratio of the scale friction to the plate-GM coefficient. For example, when predicting the performance of *C. occipitalis*, we scaled the $f_t$ function measured using the aluminum plate by 1/2.

**Table 4.** Coefficients for fourier fits used for $F_t$ in RFT calculations.

Fit function is $F_t(\beta_d) = a0 + a1\cos(w\beta_d) + b2\sin(w\beta_d) + a2\cos(w\beta_d) + b2\sin(w\beta_d)$. a0, a1, b1, a2, and b2 have units of Newtons. w is dimensionless.

| Z (mm) | a0 | a1 | b1 | a2 | b2 | W |
|---|---|---|---|---|---|---|
| 4 | 0.004041 | 0.0002925 | 0.002832 | −0.001038 | −0.0007345 | 2 |
| 6 | −0.02862 | 0.055 | −0.006713 | −0.01916 | 0.005902 | 0.7999 |
| 8 | 0.00593 | 0.01974 | −0.001129 | −0.008861 | 0.004105 | 1.232 |
| 12 | 0.01315 | 0.02947 | 0.02468 | −0.004485 | −0.005888 | 1.68 |
| 16 | 0.003115 | 0.07695 | 0.04177 | −0.01298 | −0.007372 | 1.4 |
| 20 | 0.05495 | 0.05307 | 0.05907 | −0.001921 | −0.01523 | 1.959 |
| 24 | −0.007651 | 0.1935 | 0.09582 | −0.03188 | −0.01891 | 1.359 |

The RFT calculation was carried out as follows. At each time step $\hat{n}$ and $\hat{t}$ of each of the 100 $ds$ were determined using **Equation 1**. Given some $\vec{v}_{CoM}$ and $\vec{\omega}_{CoM}$, $d\vec{F}$ can be calculated from **Equation 3** for each $ds$. Summing over the body thus gives the total force acting on the CoM in $x$ and $y$ and torque about the CoM. Assuming the animal was in the steady state, such that the total force/torque acting on the body was zero, MATLAB's lsqnonlin function was used to find the $x$ and $y$ components of $\vec{v}_{CoM}$ and (assuming planar motion) $z$ component of $\vec{\omega}_{CoM}$ which resulted in zero net force and torque at each of the 70 time steps.

## Surface granular RFT accurately predicts sand-specialist speed

We began by using RFT to estimate the steady-state $v_{CoM}$ of the sand-specialist given the average $v_{seg} = 0.67$ ms$^{-1}$ measured in experiment, snake length 39 cm, body radius 3.5 mm, and mass of 19 g. We calculated the average $v_{CoM}$ by integrating $\vec{v}_{CoM}$ over a cycle to find the total displacement of the CoM and dividing by the time to complete one cycle (the period). For scale friction equal to that of *C. occipitalis*, RFT accurately estimated $v_{CoM}$ (**Figure 7(a)**). Notably, we did not account for material hysteresis in our calculation.

Subsurface GM swimming performance was improved by reducing scale friction (**Sharpe et al., 2015**). The surface granular RFT similarly predicted that slithering speed decreased as scale friction increased (**Figure 7(a)**). This is an intuitive result, as increasing the tangential stress on the side wall by increasing friction of the scales shifts the anisotropy curve down, moving the location of $\sigma_n/\sigma_t = 1$ to larger $\beta_d$ (**Figure 6(a)**) such that force balance requires greater slipping of the body relative to the substrate and thus reduced movement speed.

At the apexes of the body wave, body segments are at $\beta_s$ near 0 deg, where $\sigma_n/\sigma_t < 1$ (**Figure 6 (a)**). These portions of the trunk experience drag without contributing thrust. Previous research found snakes moving on firm surfaces lifted these sections of the body off of the substrate (**Hu et al., 2009**). In the lab we measured 3D kinematics and found that *C. occipitalis* lifted the wave apexes such that there was a vertical wave on the body which was twice the spatial and temporal frequency of the horizontal wave (**Figure 7—figure supplement 1**). Therefore, removing these segments from contact with the material reduces drag without a decrease in propulsive force. We included lifting in the RFT calculation by assigning zero force to segments at the wave apexes (**Figure 7—figure supplement 1**).

Given the low-friction scales of *C. occipitalis*, including lifting in the calculations did not substantially improve the prediction of animal $v_{CoM}$ (**Figure 7(a)**). We speculate that these segments may be lifted as a side-effect of the muscle activation responsible for generating the horizontal waveform. It may also be that the lifting is intentional and a buffer against deleterious motor program mistakes or changes in the environment, or the small benefit of lifting these segments is greater than the energetic cost. Because the contribution of both ventral drag and lifting was minor, we did not re-distribute ventral drag forces to account for lifting.

Using the sand-specialist lifting pattern and scale friction we performed test RFT calculations using the drag stresses measured at each depth and found that the effect of depth was less than that of $\theta_m$ and less than the experimental uncertainty in $v_{CoM}$ (**Figure 7(b)**). Further, the peak of the curves did not depend on depth. Therefore, we chose to use force relations measured at a depth of 8 mm. The prediction was similarly insensitive to $v_d$, and captured the relationship between $v_{seg}$ and $v_{CoM}$ measured in the animal (**Figure 7(c)**).

Despite the observed complexity of motion at the surface–hysteresis, the potential to make and break contacts with the substrate by lifting, and speed-dependence of the drag force magnitudes–the accurate RFT predictions further served to confirm the observation that the animal was in a resistive-force-dominated regime where body inertia was negligible and performance was determined by the pattern and not the speed of self-deformation.

## Trade-off between internal and external factors constrains waveforms

While the total torque about the CoM, summed over all $ds$, at each time was zero, the torque on any given segment was not. We thus used RFT to predict the maximum joint torque, $\tau_{m,RFT}$, experienced by any *C. occipitalis* body segment over an undulation cycle (**Figure 8(a)**).

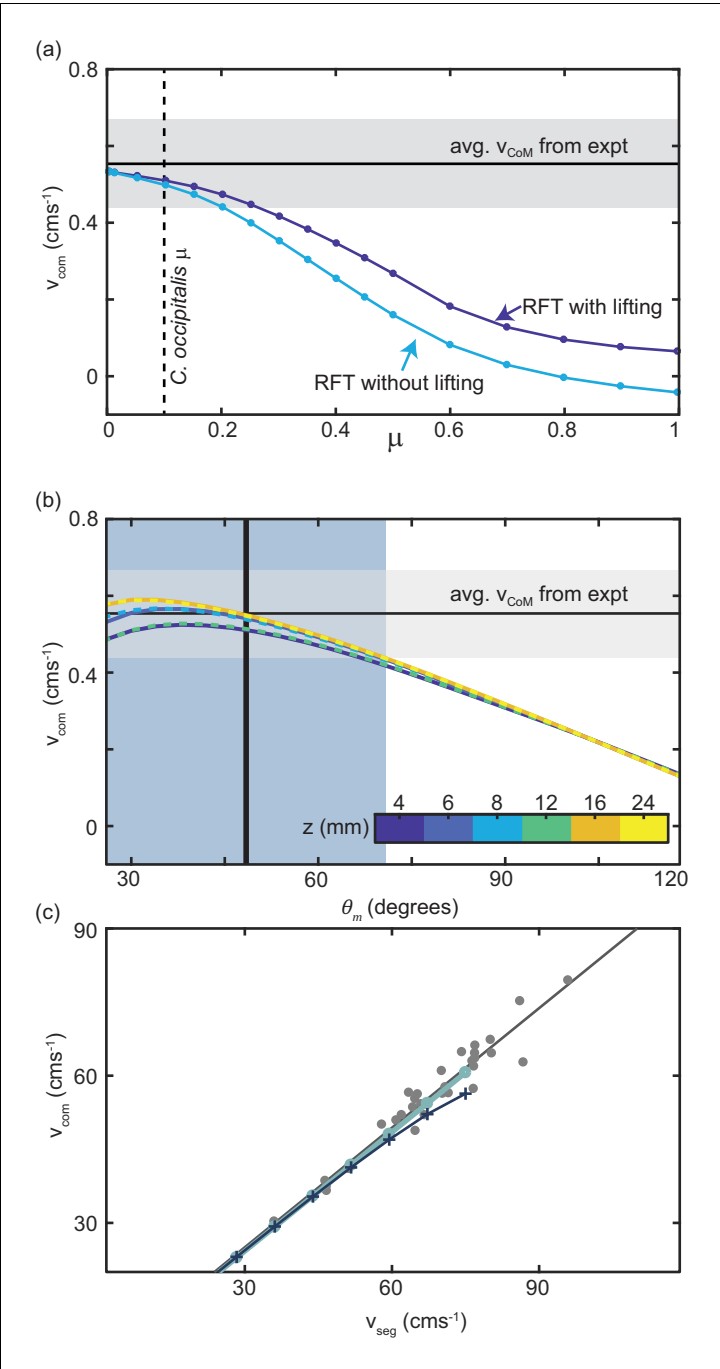

**Figure 7.** Comparison of RFT results with snake performance. (a) $v_{CoM}$ as a function of the animal scale-grain friction. $\mu = 0.1$ corresponding to *C. occipitalis* is indicated by the vertical dashed line. The light blue curve is RFT calculation for the snake parameters without any lifting and the dark blue curve was calculated for lifting of segments whose $\theta$ was in the lowest 41% (equivalent to the wave apexes *Figure 7—figure supplement 1*). The horizontal black line and gray bar are the average and range of $v_{CoM}$ calculated from experiment. (b) RFT calculation of $v_{CoM}$ as a function of $\theta_m$ using granular force relations measured at different depths, $z$, denoted by color. Three of the curves are dashed so that all six curves can be seen. The vertical line and blue bar denote the mean and range of $\theta_m$ measured in experiment. (c) Relationship between $v_{CoM}$ and $v_{seg}$ measured in *C. occipitalis* experiment (gray circles) and predicted by RFT using force relations measured at $z = 8$ mm for constant $v_d = 10$ mm s$^{-1}$ (light blue line) and $v_d = v_{seg}$ (dark blue line and cross markers). Solid gray line is a linear fit to the animal data $v_{CoM} = 0.82 v_{seg} + 0.54$ (R-square = 0.94).

The online version of this article includes the following figure supplement(s) for figure 7:

*Figure 7 continued on next page*

*Figure 7 continued*

**Figure supplement 1.** Measurement of the vertical wave of lifting.

We calculated torque acting on the CoM assuming infinitesimal width and finite segment length. For joint $i$ the internal torque was $\tau_{i,int} = \sum_{k=1}^{i-1} (\vec{r}_k - \vec{r}_i) \times d\vec{F}_k$ (*Ding et al., 2011*). We find this value for each of the segments at all times. $\tau_{m,RFT}$ is then the maximum value in this $100 \times 70$ matrix.

$\tau_{m,RFT}$ increased as $\theta_m$ or $\xi$ decreased. This was both because balancing forces required larger $\beta_d$, thus greater force magnitudes (*Figure 5(b)*), and those shapes 'stretched out' the body, creating longer lever arms (see *Figure 8(b)* and compare shapes in lower left corner to upper right).

The snakes could minimize torque by increasing $\xi$ and/or $\theta_m$, however, we did not observe these waveforms (*Figure 2*). Similarly, snakes did not use waveforms which minimized mechanical cost of transport or distance traveled per cycle (*Figure 8—figure supplement 1*). We thus hypothesized there were internal factors which 'penalized' the low-torque waveforms.

If, for simplicity, we assume the animal moved with no slip regardless of waveform such that the distance traveled per cycle was equal to the wavelength, there were two (not necessarily independent) ways to increase $v_{CoM}$: increase the wavelength by decreasing $\xi$ and/or $\theta_m$ (see example waveshapes in *Figure 8(b)*) or increase the speed of wave propagation (given the linear relationship between propagation speed and $v_{CoM}$, *Figure 7(c)*).

In limbed organisms, movement speed is related to the interplay between gait parameters like stride length and frequency and physiological concerns like energetic cost (e.g. *Hoyt et al., 2006*; *Kram and Taylor, 1990*). We hypothesized that, because the snake performed an escape response in our experiments, their objective was to maximize speed.

For a desired $v_{CoM}$, we explored how quickly a 'muscle' segment of fixed nominal length relative to total body length would have to shorten as a function of $\xi$ and $\theta_m$. The shortening speed was a function of both how quickly curvature changed along the body for a given waveshape and the frequency needed to achieve the target $v_{CoM}$ for that shape's stride length. We modeled a body segment as a bending beam with length along the spine $\delta s$ and arclength along the inside of the curve $\delta s'$ (see diagram *Figure 8(b)*). We approximated the amount of shortening the muscles must produce as the difference, $\Delta s = \delta s - \delta s'$, at the point of maximum bending. Using this model we

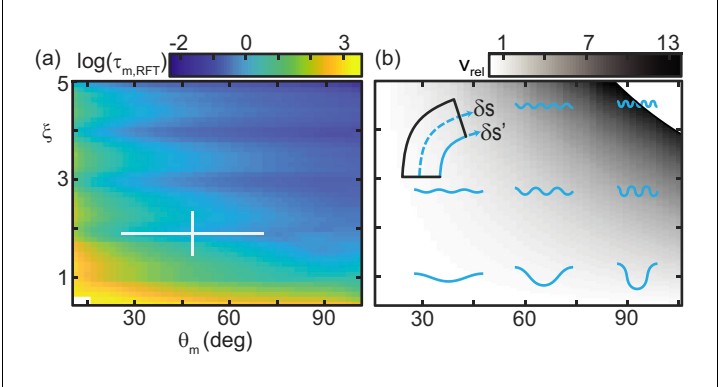

**Figure 8.** Trade-off between external torque and internal actuation speed. (a) RFT prediction of the peak joint level torque, $\tau_{m,RFT}$, occurring during one undulation cycle as a function of $\theta_m$ and $\xi$. Color indicates the log of $\tau_{m,RFT}$. White cross is the mean and range of animal $\theta_m$ and $\xi$. measured in experiment. (b) Heatmap of relative actuation speed, $v_{rel} = \frac{v_{short,o} - v_{short}(\theta_m,\xi,f)}{v_{short,o}}$ at the nominal waveform of *C. occipitalis* ($\theta_m = 48.4$, $\xi = 1.9$). Frequency, $f$, is calculated to yield the desired $v_{CoM}$ for a given waveform's stride length (see Materials and methods for details). Diagram in upper left corner illustrates the bending beam. The nominal length of a segment at the midline is $\delta s$, blue dashed line. The length of the inside of a segment is $\delta s'$, solid blue line. Relative actuation speed is the rate of change of $\delta s'$ at the nominal $v_{CoM}$, assuming no-slip motion.

The online version of this article includes the following figure supplement(s) for figure 8:

**Figure supplement 1.** Surface RFT results for mCoT, BLC, and $\tau_{m,RFT}$.

estimated $v_{shorten} = \delta s / \frac{1}{4} T$, the speed the inside of the bend must change length to go from $\delta s' = \delta s$ at the inflection points of the wave to the maximum $\delta s'$ occurring at the peaks given undulation period $T$ set by the frequency (see Materials and methods for details).

Figure 8(b) is a heatmap of the increase or decrease in the required shortening speed relative to a nominal shortening speed, $v_{short,o}$ (which we take to be that occurring at the mean $\theta_m$ and $\xi$ measured in experiment). The actuator speed increased with $\xi$ and $\theta_m$. The reason for this was twofold. Firstly, both greater amplitudes and wavenumbers required each segment to enact a greater change in body curvature. Secondly, increasing $\xi$ and/or $\theta_m$ decreased stride length, requiring greater frequency to maintain CoM speed. This is somewhat counter intuitive when considering (non-slip) limbless undulatory gaits as compared to limbed gaits. Increasing stride length on a limbed locomotor requires a greater amount of actuation as, e.g., the hip joint must sweep out a larger angle. Therefore to maintain a CoM speed there is a trade off between increasing/decreasing stride length and decreasing/increasing frequency. In contrast, we see here that actuation speed monotonically decreases as the wavelength increases; if we consider only actuation speed, the snakes' waveforms are under-performing.

This result rationalized why all wave parameters we measured inhabited the center of the wave parameter space (Figure 2); there was a trade off between external forces and internal actuation constraints.

## C. occipitalis waveform maximizes movement speed under anatomical constraint

We endeavored to include the trade off between decreasing torque and increasing actuation speed needed to maintain $v_{CoM}$ (now using RFT to estimate the actual stride length given slipping) as $\theta_m$ and $\xi$ increased. To do so, we used RFT to calculate the joint-level power of each segment over a cycle for each shape in the $\theta_m$, $\xi$ space.

The angle between adjacent segments, $\zeta$, relates to $\theta$ via $\zeta = \frac{d\theta}{ds}$. Joint-level power for each joint at each time was calculated using the rate of change of the joint angle, $P_{i,int} = \vec{\tau}_{i,int} \cdot \frac{d\vec{\zeta}_i}{dt}$, where

$$\frac{d\zeta_i}{dt} = \frac{d^2\theta_i}{dtds} = -\theta_m \left(\frac{2\pi\xi}{L}\right)^2 v_{seg} \sin\left(\frac{2\pi\xi}{L}(s_i + v_{seg}t)\right). \tag{5}$$

The power-limited velocity, $v_{pl}$, was the CoM speed for which the peak power generated by any joint over the cycle was equal to a constant peak available power.

In accord with the tradeoff between internal actuation demands and external torques, $v_{pl}$ was maximized in the center of the $\theta_m$, $\xi$ space inhabited by the waveform of C. occipitalis (Figure 9(a)). Reflecting the oscillations in $\tau$, this metric had maxima near integer values of $\xi$ (Figure 9(a,b)). Power-limited velocity was maximized at the number of waves (Figure 9(b)) as well as at the attack angle (Figure 9(c)) used by C. occipitalis. We estimated the peak torque output of C. occipitalis' muscles using dissection of museum specimens (see Materials and methods for details). The attack angle used by the animals was near the point where $\tau_{m,RFT}$ increased above the estimated muscle capability. We report both maximal torque output (Figure 9(c) upper horizontal bar) as well as output scaled for the average contraction velocity estimated as the body-wall shortening speed (Figure 9(c) lower horizontal bar). It may be that the individual variation in $\theta_m$ reflects differences in peak muscle power capabilities.

## Stout snakes must use larger attack angles to succeed

Our granular RFT calculations rationalized the stereotyped waveform used by C. occipitalis. The waves of the non-specialists, however, displayed more variation (Figure 2(c)). Using the insights and RFT calculation we developed in our study of the sand-specialist we endeavored to understand how the differences in waveform and morphology impacted performance.

A striking difference between the various species was their aspect ratio, $A = L/w$, the ratio of the total length to the diameter of the body at the widest point. A slender snake like Figure 1(b,c) will thus have a higher aspect ratio than a stout one, for example Figure 1(a). We previously discovered that C. occipitalis' high-aspect-ratio allowed it to move more effectively when swimming subsurface in GM than a low-aspect-ratio sand-swimming lizard (Sharpe et al., 2015).

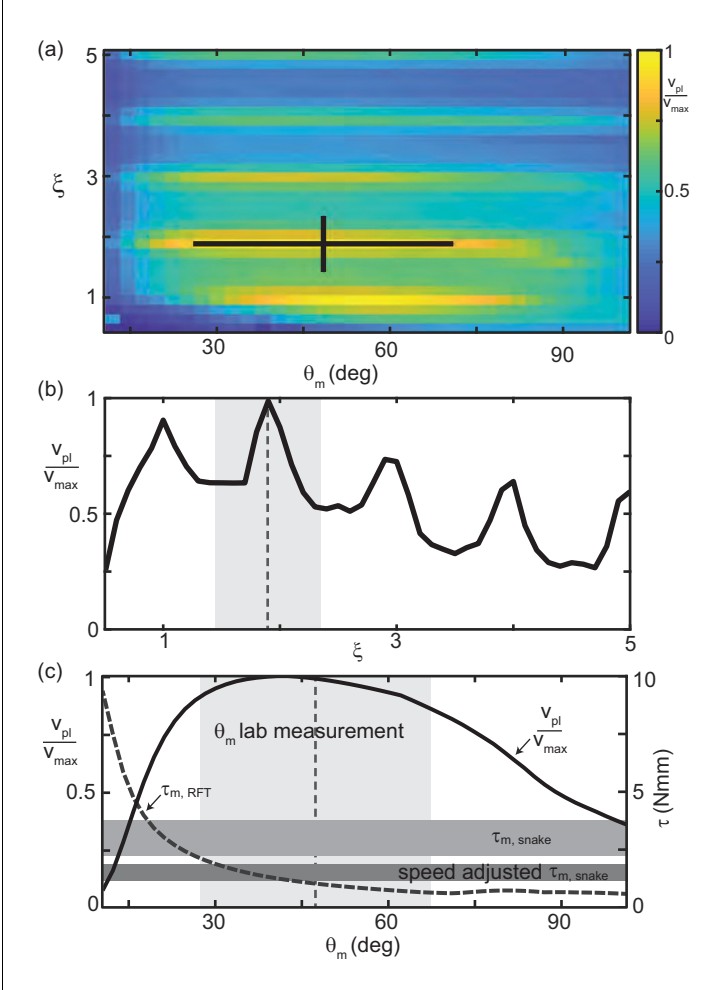

**Figure 9.** RFT reveals waveform balances internal and external concerns in *C. occipitalis.* (a) Segment-power-limited velocity, $v_{pl}$, divided by $v_{max}$, the largest value of $v_{pl}$, in the $\theta_m$, $\xi$ space. Black cross is the snake waveform as in (*Figure 2(c)*). (b) $v_{pl}/v_{max}$ versus $\xi$ for $\theta_m = 48.4$ deg, the average experimental value. This is a vertical slice of (a). The vertical dashed line and gray shaded area are the mean and range of snake $\xi$ values. (c) Solid black curve $v_{pl}/v_{max}$ (left vertical axis) and gray dashed curve $\tau_{m,RFT}$ (right vertical axis) versus $\theta_m$ for $\xi = 1.90$ (the average snake value) are horizontal slices of (a) and *Figure 8(a)*, respectively. The vertical dashed line and light gray shaded area are the mean and range of snake $\theta_m$. Horizontal gray bars are the estimated muscle torque capabilities of the snakes (see Materials and methods for details). The upper bar, $\tau_{m,snake}$ is the estimated maximum muscle torque and the bottom bar, speed adjusted $\tau_{m,snake}$, is the maximum torque reduced to reflect the average muscle shortening speed of the snakes at the average experimentally measured velocity.

We characterized the performance of the snakes using the unsigned average slip angle, $\beta_s = \mathrm{acos}(|\hat{t} \cdot \hat{v}|)$ (*Figure 3(a)*, inset; *Sharpe et al., 2015*). For each trial we calculated slip along the body at each moment in time and calculated the mean and standard deviation of these values. This variable captures how much the movement of the animal is as if it were moving 'in a tube'. If the slip is zero we expect each segment follows in the path of the segment before it whereas when slip is large the body is primarily sliding laterally to the midline.

We used RFT to predict the relationship between aspect ratio and slip value. We used the average mass m = 0.76 kg, length L = 0.89 m, $\theta_m = 48$, and $\xi = 2.38$ of all non-sand-specialist snakes with slip < 30 deg and assumed constant body segment length (i.e. all snakes divided into same number of segments), constant body density, $\rho = 1000$ kgm$^{-3}$, and a circular cross section. The width and length of a snake can then be calculated as $w = (\frac{m}{\pi \rho A})^{\frac{1}{3}}$ and $L = wA$.

In modeling the various snake species we used frictions 0.1, 0.15, and 0.2. This represents a reasonable range of the friction coefficient between forward-sliding ventral scutes and the substrate during lateral undulation (*Hu et al., 2009*; *Baum et al., 2014*), although we note that higher coefficients have been measured on rough material and can be actively modulated by the animal (*Marvi and Hu, 2012*). We did not account for a difference in intrusion depth which may occur as mass increased. This relationship likely depends on both pressure applied to the GM as well as its properties.

For an animal with fixed mass and internal density, RFT predicted a modest increase in slip with decreasing $L/w$ (*Figure 10(a)*). Increasing/decreasing scale friction shifted this curve upwards/downwards (*Figure 10(a)*, gray patch). This inverse relationship between aspect ratio and slip was also measured in the animals (*Figure 10(a)*, circle markers).

There was a transition in the slip versus aspect ratio plot at an aspect ratio of about 26 (*Figure 10(a)*, vertical dashed line). At larger $L/w$ the snakes were, with the exclusion of one exception, moving effectively with an average $\beta_s = 10.3 \pm 3.8$ deg (*Figure 10(a)*, blue crosses). Below $L/w = 26$ performance was highly variable. Four of the five species which failed to progress (wave efficiency less than 0.35, *Figure 10—figure supplement 1*) across the level sand had $L/w<26$ (*Figure 10(a)*, red crosses). The other snakes of $L/w<26$ moved with higher slip on average than their high-aspect ratio counterparts, $\beta_s = 20.3 \pm 5.8$ deg (*Figure 10(a)*, gray crosses).

We found that animals with $\beta_s>30$ degrees were those that were ineffective (*Figure 10(a)*, red crosses, *Figure 10—figure supplement 1*). These animals either did not progress at all (e.g. *Figure 1 (a,d)*), or would have only occasional spurts of forward motion linked by extended periods of undulating in place. Even snakes close to, but less than, 30 deg were able to make consistent forward progress (e.g. *Figure 1(e)*).

RFT predicted that as attack angle decreased, the slip of a snake with fixed body morphology and wavenumber would increase (*Figure 10(b)*, black line). We examined slip as $\theta_m$ changed for those snakes with an aspect ratio less than 26 (and the snake that failed with $L/w$=30). Those which failed to progress (*Figure 10(b)* red crosses) were at lower attack angles than those which succeeded (*Figure 10(b)*, black crosses)(median and std. of all values $\theta_m(\beta_s \geq 30) = 33.2 \pm 21.3$ N = 6, n = 4311 measurements, $\theta_m(\beta_s<30, L/w<26) = 48.8 \pm 24.5$ N = 12, n = 9017, p << 0.01). Those snakes which had $L/w$ of greater than 26 generally used higher attack angles as well (*Figure 10* (b, inset) blue crosses).

We used RFT to calculate the performance of individual snakes using that animal's mass and length and its average wavenumber and attack angle (select points shown in *Figure 10(a)*, square markers). For those snakes with $\beta_s<30$ deg and $L/w>26$ the average difference between the measured and predicted values was 5.1 deg (for individuals with more than one trial we compared to the mean of the average slip values, *Figure 10(a)*, black squares). For those with $L/w<26$ and slip <30 the calculation was less accurate; the average unsigned difference was 15.3 deg (*Figure 10(a)*, black squares). RFT calculation was the least accurate, average difference 29.9 deg, for those snakes which failed ($\beta_s>30$ deg, *Figure 10(a)*, red squares). The accuracy of the RFT estimation was inversely correlated with the amount of slip (*Figure 10—figure supplement 2*).

We observed that the snakes created permanent disturbances in the surface of the GM. Those which moved with low slip created observable piles of sand at the posterior-facing side/s of the body (*Figure 1(b,c)*), but would not push these piles so far that they would collide with previously-created mounds. In contrast, those which failed appeared to dig themselves into a channel, primarily depositing GM laterally to the long-axis of the body (*Figure 1(a)*). We noticed that with each undulation these animals would create new piles and then push them until they contacted the mounds of material left behind by previous wave cycles. These material re-interactions occurring during high slip locomotion were not included in the RFT calculations; we hypothesized that once slip was substantial enough that disturbed material was re-encountered in subsequent undulations the material memory impacted performance.

## Material remodeling regulates locomotion

The waveform and performance of the snakes was variable, and this variability was reflected in the tracks left by the animals. As the material did not re-flow around the body after being disturbed, we hypothesized that the manner in which different waveforms remodeled the substrate was important

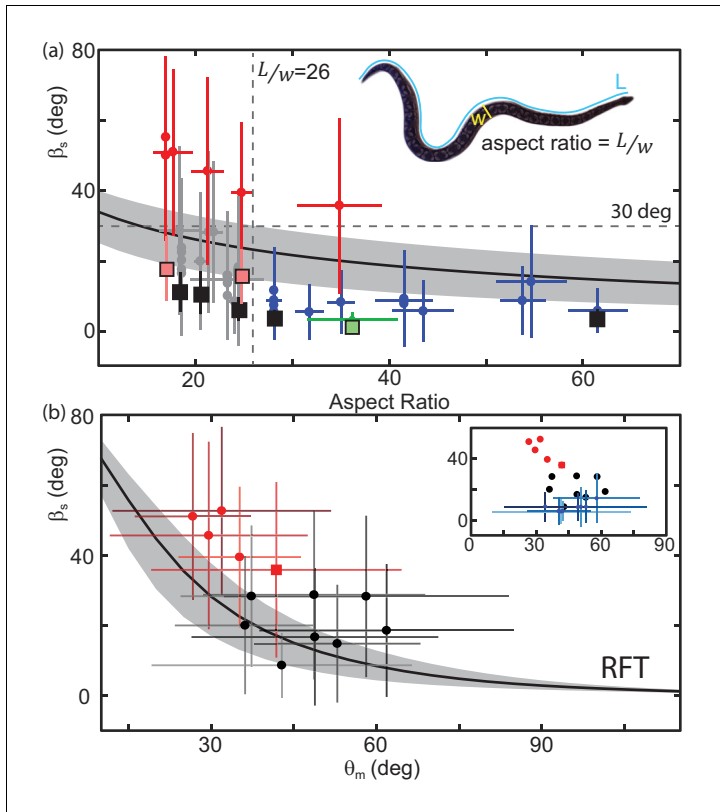

**Figure 10.** Performance depends on aspect ratio and waveform. (a) Slip as a function of aspect ratio, $L/w$ as illustrated on the snapshot of a snake (*Acrantophis dumerili*, $L/w = 23.2$) where $L$ is the total length of the snake (blue curve) and $w$ is the width at the widest point (yellow line). Circle markers and vertical lines indicate the mean and std. of each trial. Horizontal bars are the range of aspect ratios measured from video stills by two different researchers. Black are successful trials, red are failures, green is *C. occipitalis*. Each animal tested had a unique aspect ratio; for the non-sand-specialists, multiple data points at the same $L/w$ indicate multiple trials for the same animal. N = 22 animals, n = 38 trials. Only mean shown for *C. occipitalis* (N = 9, n = 30). Black curve is the RFT estimation of slip using a scale friction of $\mu = 0.15$. Gray area indicates predictions for $\mu = 0.1$ (lower slip) and 0.2 (higher slip). Square markers indicate RFT prediction of slip for that animal using a scale friction of $\mu = 0.15$ with vertical lines showing range from $\mu = 0.1$ to 0.2. For *C. occipitalis* $\mu = 0.1$ (*Sharpe et al., 2015*) and min/max = 0.05/0.15. All RFT predictions used force relations for 300 μm glass beads assuming intrusion depth of 8 mm. (b) Slip versus $\theta_m$ for animals with $L/w<26$. Mean and standard deviation of all measurements in a trial. Square marker is the snake $L/w>26$ which failed. If an animal performed more than one trial we combined the measurements of all trials before taking the mean and std. dev. Successful trials in black, failures in red. Black curve is the RFT prediction using the average waveform and anatomy values calculated using all species studied, $\xi = 2.5$, mass = 0.63 kg, L = 89 cm, and an estimated scale friction of $\mu = 0.15$. Gray band indicates predictions for friction $\mu = 0.1$ and 0.2. (inset) Black and red circles are the averages from the main plot. Blue markers are successful trials $L/w>26$.

The online version of this article includes the following figure supplement(s) for figure 10:

**Figure supplement 1.** Wave efficiency versus slip.

**Figure supplement 2.** Slip measured in experiment versus RFT prediction.

in determining performance. We systematically explored the impact of waveform on performance using a robophysical model, a 10 joint robot on the surface of poppy seeds (*Figure 11*).

## Robophysical model elucidates connection between waveform and performance

The robot executed serpenoid curves (*Equation 1*), realized using a Lynxmotion SSC-32 servo controller to dictate the angles between adjacent motors, $\zeta$, (*Figure 11(a)*). Each motor was offset from

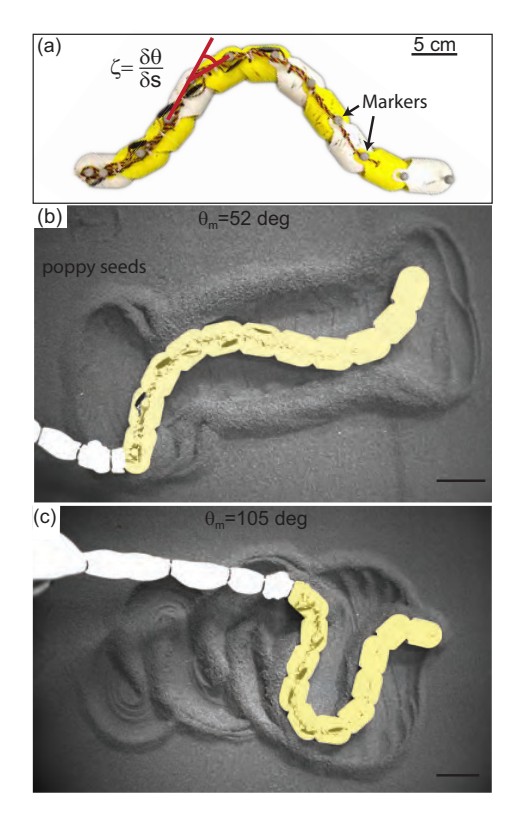

**Figure 11.** Systematic exploration of substrate memory effects using a robophysical model. (**a**) The 10 joint, 11 segment robot constructed using ten Dynamixel AX-12A servo actuators (Robotis) with a stall torque of 15.3 kg·cm. Yellow and white segments are 3D printed cylindrical casings, each housing one motor. Length = 72 cm and $L/w = 11.3$. (**b,c**) Stills of the robot taken after it has completed 2.75 undulation cycles. $\theta_m = 52$ deg in (**b**), $\theta_m = 105$ deg in (**c**), and $\xi = 1$ in both cases. Robot is artificially colored yellow to distinguish the body from the tail cord in white. Note the large appearance of the tail cord in c is due to perspective as the researcher was standing and loosely holding the cord above the substrate.

its anterior neighbor by a constant phase set by $\xi$. We varied $\xi$ and $\theta_m$ and recorded kinematics as the robot performed three undulation cycles on the surface of poppy seeds in a $1.2 \times 1.8$ m$^2$ bed. The GM was manually smoothed between trials. Four OptiTrack cameras (Flex 13, Natural Point) tracked infrared reflective markers on the robot (*Figure 11(a)*). The number of joints on the robot was limited by motor strength. Thus, the robot could only achieve $\xi$ between 1 and 1.4 because lower $\xi$ required more torque than the motors could provide and higher were not well-resolved by the number of joints. The power per unit volume of the motors also limited the aspect ratio; $L/w = 11.3$ is lower than that of the stoutest snakes tested.

We characterized performance by measuring slip as in the snakes (*Figure 12(a)*). We also measured the CoM displacement in a single undulation cycle normalized by the total length (BLC, *Figure 12(b)*). This variable provided intuition for how effectively the robot was progressing across the substrate.

Performance of the robot was a function of the waveform parameters. This was intuitive given our RFT predictions for the snake, which depended on both $\xi$ and $\theta_m$ (e.g. *Figure 9*), and was commensurate with our findings in the biological experiments (*Figure 10 (b)*). Over the range of $\xi$ available to the robot, performance was more strongly dependent on attack angle than wavenumber, therefore we focused our attention on variations in $\theta_m$.

## RFT prediction of robot performance is inaccurate

We used granular stress relations measured in poppy seeds using a flat plate of the same plastic used to print the robot body (*McInroe et al., 2016*) in a resistive force theory calculation predicting the robot performance as a function of $\theta_m$ for fixed $\xi = 1$ (*Figure 12 (a,b)*, black curves).

In previous studies, RFT accurately predicted locomotor performance of a number of systems (*Zhang and Goldman, 2014*) and in our study RFT was accurate for *C. occipitalis* as well as the successful non-sand-specialist snakes. However, as in the high-slip snakes, RFT under-predicted slip at low attack angles (*Figure 12 (a)*). Despite the substantial predicted slip, RFT indicated these waveforms would still make forward progress (*Figure 12 (b,e)*). The robot, however, did not displace at the smallest attack angles tested (*Figure 12 (b,c)*). Conversely, at high $\theta_m$ RFT under-predicted displacement of the robot (*Figure 12 (b)*).

At both low and high attack angles RFT over-predicted the amount of yaw (rotation about the CoM in the horizontal plane) that would occur (*Figure 12* (c:f)). The discrepancy was likely because this calculation assumed the robot was always encountering undisturbed material. This assumption worked for subsurface swimming where the GM behaves like a frictional fluid, constantly re-flowing to fill in the spaces cleared by motion of the body. We observed that the motion of the robot

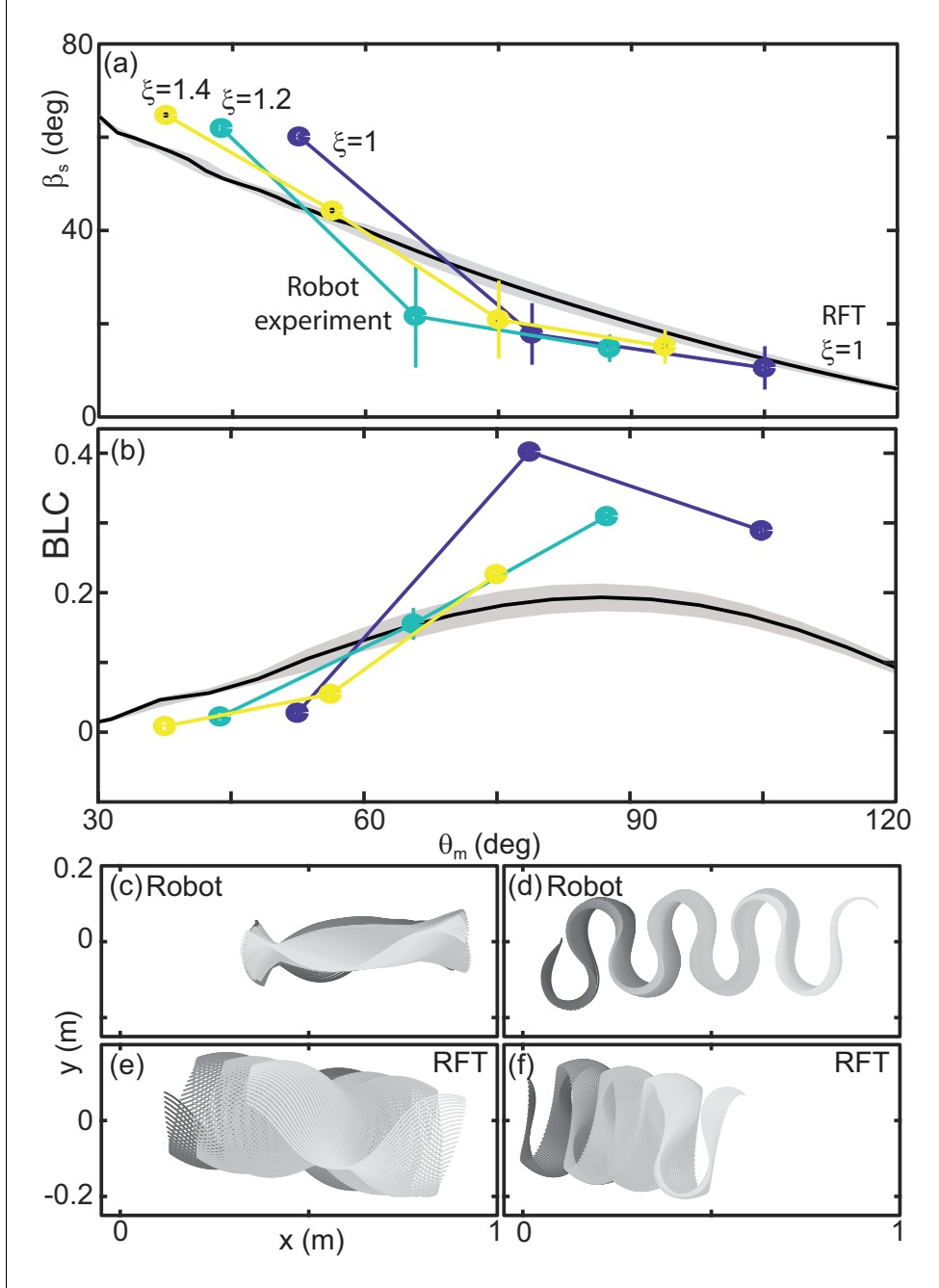

**Figure 12.** Robot performance is sensitive to the waveform and poorly predicted by RFT. (**a**) Slip versus $\theta_m$. Robot measurements are circles connected by solid lines. $\xi$ is indicated by color, yellow is 1.4, teal is 1.2, and blue is 1. Mean and std. of three trials. Where error bars are not visible they are smaller than the marker. RFT prediction is the solid black curve. The gray shaded region indicates uncertainty in our knowledge of the robot's mass (because of the weight of the tail cord which is held above the substrate) and friction coefficient. (**b**) Body lengths traveled in a single undulation cycle (BLC) as a function of attack angle. Colors and lines are consistent with (**a**). (**c,d**) Robot and (**e,f**) RFT predicted kinematics for $\xi = 1$. Color indicates time from the beginning to the end (darker to lighter gray). Both robot and RFT trajectories include three complete undulations. (**c,e**) $\theta_m = 52$ deg. (**d,f**) $\theta_m = 105$ deg.

deposited some material lateral to the direction of motion. These piles, when re-encountered, appeared to limit yaw. We posited that material hysteresis played an important role in performance.

## Material remodeling changes performance

We observed that material flow was a function of the robot waveform. We recorded overhead video at 120 fps (AOS S-Motion, Baden Daettwil, Switzerland) and estimated grain velocities using particle image velocimetry (PIV, PIVlab ; *Thielicke and Stamhuis, 2014a*; *Thielicke and Thielicke, 2014b*). We characterized the overall structure of the flow by measuring $\psi_v$, the angle between the velocity vector of the grains at a given point in time and space and the overall direction of motion of the robot (*Figure 13(a)*). We calculated the probability density to measure a given $\psi_v$ over a run for $\xi = 1$ and each of the three $\theta_m$ tested (*Figure 13(b)*).

At high attack angles significant amounts of the material were deposited by the posterior-facing body segments, similar to what we observed in the successful snakes (*Figure 11(c)*, *Figure 13— video 1*). Consistent with this observation, we measured peaks in the probability density of $\psi_v$ near ± 180 deg, corresponding to grains which are flowing opposite the average direction of motion of the robot (*Figure 13(b)*, orange curve). When the granular piles moving with the body at later points in time re-encountered these piles the additional stress resisted backward slipping of the robot, increasing displacement above that predicted for a frictional fluid (*Figure 12(b)*).

The GM being pushed by low $\theta_m$ waveforms had a significant velocity component lateral to the direction of motion; as $\theta_m$ decreased, the peaks in the $\psi_v$ probability density shifted toward ±90 deg (grain flow perpendicular to the average direction of motion, *Figure 13(b)*, green and blue curves). These piles were thus deposited to the side of the robot and with each undulation more material was added to these 'sidewalls' which can be seen in *Figure 11(b)*. As the GM being pushed by subsequent undulations encountered these pre-existing walls the robot was apparently unable to overcome the inertia of the grains in the previously created piles, instead depositing more material in the walls without changing their location. The result was that the robot swept out a trough in the GM, completely stalling forward progress and sometimes even moving backward as the trunk 'rolled' along the trough walls (*Figure 13—video 1*). These failures were similar in appearance to those observed in the living snakes (*Figure 1(a,d)*, *Figure 13—video 2*).

In limbed systems, RFT has often been useful in predicting the performance of the first gait cycle, before material re-interaction could occur (*Mazouchova et al., 2013*; *McInroe et al., 2016*). In our limbless robot, however, RFT was incorrect even for the first cycle (*Figure 13(c)*). Enhancement or degradation of the performance was occurring within the first gait cycle as portions of the body contacted GM disturbed by other segments at a previous time.

This provides some intuition as to why performance of *C. occipitalis* was robust to variations in the waveform which would have impacted the robot (*Figure 2*); because this animal's long body and slick scales yield low-slip slithering, the body and/or piles of GM being pushed by the body are continuously encountering new material through time. The animal is effectively moving in a frictional fluid even though the material is not re-flowing as in subsurface movement.

This raises an interesting subtlety: the tail is always moving through material disturbed by the head, and placing a successful robot waveform back in its own tracks does not decrease performance (*Figure 13—video 3*). This underlines the idea that locomotive failure does not arise solely because the material is not pristine. Rather, it is the interdependence between the motion of the animal or robot and the evolving material state, both of which depend on the history of the system, that leads to either beneficial or deleterious reaction forces. Despite the complicated interplay between the robot and the substrate, we found the resulting performance was repeatable. For example, for all three attack angles tested at $\xi = 1$, each of the three trials for a given waveform pushed material in a similar way and displaced a similar amount in each of the three undulations (*Figure 13(b,c)*). This indicated that, while the material remodeling was complex, it was deterministic.

Studies found that generalist snakes chose self-deformations in response to available environmental forces (*Gray and Lissmann, 1950*, *Kelly et al., 1997*), a strategy which we might expect would be useful on challenging, deformable materials like GM. However, we found evidence that C. occipitalis did not change its waveform when faced with changes to the surroundings (*Schiebel et al., 2019*) and our robot was executing predetermined waveforms. The ability to sense and respond to induced changes in the substrate is apparently unnecessary given the initial selection of an appropriate pattern of self-deformation.

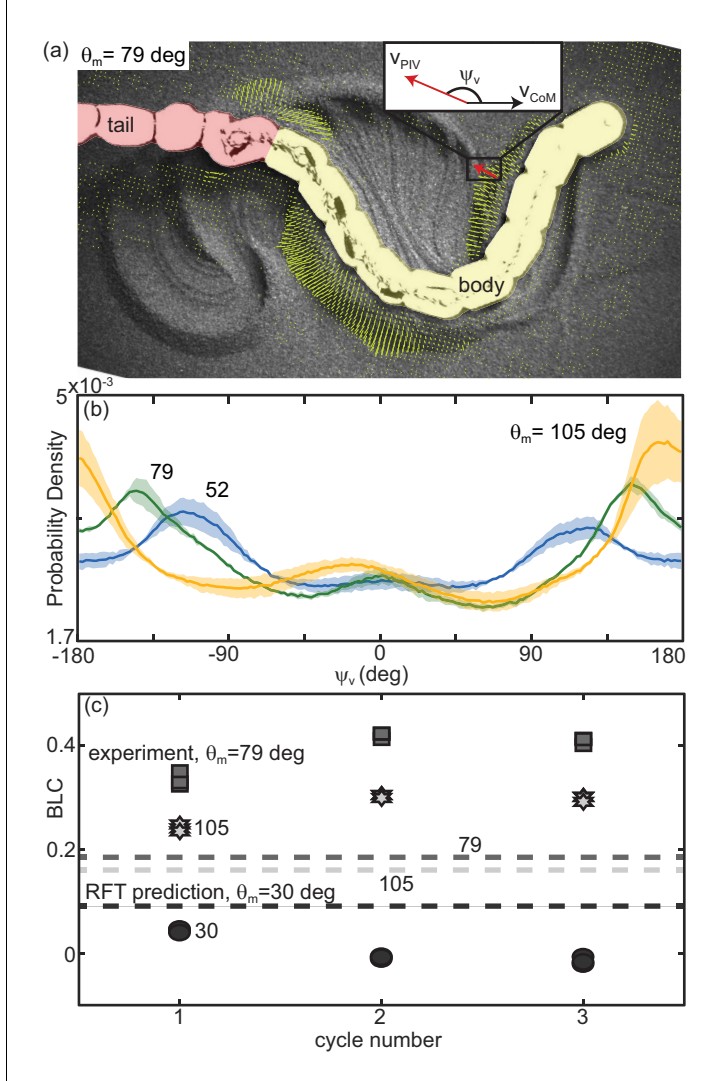

**Figure 13.** Material remodeling determines kinematics of the robot. (a) Snapshot of the robot moving on poppy seeds with grain velocity vectors estimated using PIV shown in yellow. We measured the grain velocity vector angle, $\psi_v$ as shown. A mask of the body (yellow) and tail (red) was used in all PIV calculations and vectors both too close and too far from the body were not included in calculating $\psi_v$. Vectors shown are representative of those included. We collected three trials per condition and three complete undulation cycles per trial. (*inset*) Diagram of $\psi_v$ for an example estimated grain velocity vector in red. $v_{CoM}$ is pointing in the average direction of motion of the robot. (b) Probability density of $\psi_v$ for each of the three $\theta_m$ tested, $\theta_m = 105$ (yellow), 79 (green), and 52 (blue) as labelled on the plot. Curves were calculated using all $\psi_v$ measured in space and time in a single trial. Solid lines are the average curve and the shaded area indicates the standard deviation across the three trials. (c) Distance traveled by the robot in a single step, normalized by body length, measured at the end of each of three consecutive undulation cycles. $\xi = 1$ for all trials shown. From darkest to lightest gray color: circles are $\theta_m = 30$ deg, square are 70 deg, and stars are 105 deg. Three trials are plotted separately for each step. RFT prediction shown as a horizontal lines with color corresponding to experiment.

The online version of this article includes the following video(s) for figure 13:

**Figure 13—video 1.** Snake robot moving on poppy seeds.
https://elifesciences.org/articles/51412#fig13video1
**Figure 13—video 2.** Robot and snake failure comparison.
https://elifesciences.org/articles/51412#fig13video2
**Figure 13—video 3.** Snake robot replaced in old tracks.
https://elifesciences.org/articles/51412#fig13video3

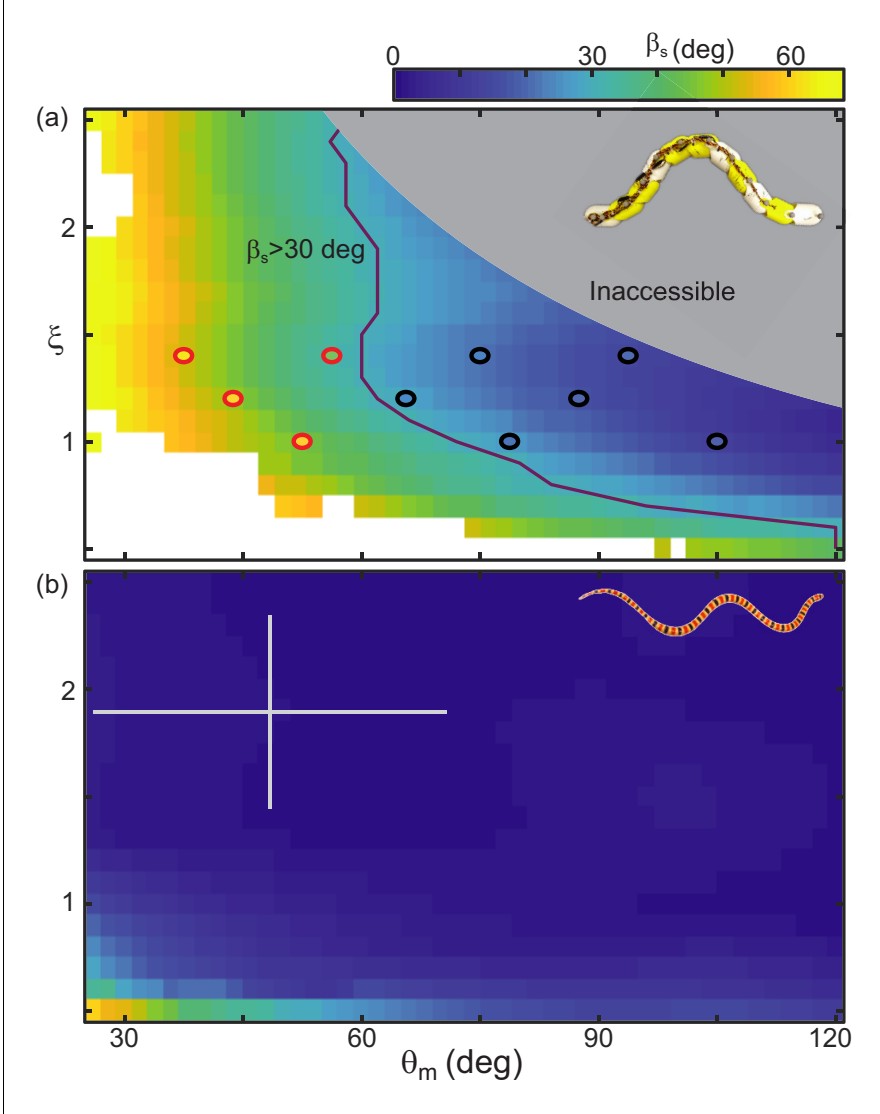

**Figure 14.** RFT prediction of slip gives a heuristic for performance. (a) Color denotes slip as $\theta_m$ and $\xi$ vary, calculated using RFT for robot dimensions and poppy seed force relations. Circles indicate the parameter combinations tested on the robot. Waveforms which were successful over three undulations are black, red markers denote unsuccessful parameters (distance per undulation over wavelength <0.35, **Figure 10—figure supplement 1**). Purple line indicates where RFT predicts $\beta_s \approx 30$ deg. The lightly shaded region denotes waveforms which the robot cannot perform. White areas indicate where the RFT calculation did not converge on a solution for at least half of the time steps. (b) Slip calculated for *C. occipitalis* morphology moving on 300 µm glass particles. Color is the same as in (a). White cross denotes mean and range of trial averages from sand-specialist experiments.

## RFT predicts performance transition

Although RFT did not consistently predict the robot's slip, it did provide a heuristic for success and give some intuition as to why a locomotor with the morphology of the robot is challenged by GM. The line in the $\theta_m$, $\xi$ space which RFT predicted to be at $\beta_s \approx 30$ deg separated the successful and unsuccessful robot waveforms (**Figure 14(a)**).

We found there were a limited range of waveforms predicted to both have $\beta_s \leq 30$ deg and be accessible to the robot given its aspect ratio and joint resolution (**Figure 14(a)**). In contrast, *C. occipitalis* had a large space of parameters for which slip was small (**Figure 14(b)**). This suggests that, in contrast to the robot which was sensitive to waveform, the snake was robust to fluctuations in the motor pattern.

The snake also had the benefit of greater resolution than the robot; the animal has $\approx$ 120 vertebrae (note that in snakes there is variation in vertebral number between species and between individuals of the same species) while the robot has only 10 joints. This flexibility allowed the snake to operate well away from low-performing shapes (low $\xi$ and $\theta_m$). Increasing resolution would allow the robot to use more waves on the body, making a greater range of low-slip waveforms available (those in the upper right-hand corner of *Figure 14(a)*). However, the robot is limited by the size and strength of the motors. Adding more actuators would increase torque on the central joints, preventing accurate realization of the commanded shapes.

## Conclusions

Undulatory motion on the surface of materials with memory is challenging for both animals and robots because the sensitive connection between a locomotor's waveform and its performance is complicated by material remodeling. We begin the search for principles of locomotion on substrates with memory by studying a sand-specialist snake, generalist snakes, and a robophysical model. We also used new granular drag measurements to extend resistive force theory to calculate surface performance. By comparing the limbless locomotors and using surface RFT calculations, we identified a number of factors which improved undulatory locomotion on our hysteretic material—sufficiently high attack angle waveform, high aspect ratio body, low scale friction, and lifting of the wave apexes. The generalist snake and robot experiments demonstrated that possessing all of these is not necessary for motion. However, as morphology becomes less optimal (more stout, higher friction) effective management of the substrate remodeling via choice of the appropriate waveform was increasingly crucial to progress.

The interrelationship between morphology, waveform, terrain remodeling, and performance rationalizes why it has been difficult to develop slithering robots which move on terrestrial materials with memory as compared to swimming in fluids or using rigid obstacles. Currently, robots are limited by the power density of the motors to low aspect ratio morphologies with a limited range of available waveforms such that they are in a regime where small changes in waveform can impact progress through the environment. The continued development of smaller, high-power-density motors will allow snake-like robots with higher aspect ratios and more joints. This study indicates that such robots will correspondingly be more adept at movement in substrates with memory, like those found in many natural terrains.

An immersed locomotor is aided by fluid-like re-flowing of the GM which mitigates memory-dependent failure modes (*Maladen et al., 2011a*). However, when *C. occipitalis* is on the surface its escape response is to flee across the GM rather than dive into it (*Warren, 1953*). We hypothesize this is because, despite the challenges we identified, there are a number of advantages for a locomotor on the surface. The granular stress magnitudes are less than subsurface and granular stress anisotropy rises more sharply such that the faster-moving, low attack angle shapes can be effective. Drag forces are reduced by avoiding the need to push the head through the material and the opportunity to lift segments which are not generating thrust. There is also less friction on the body, which at the surface is due only to animal mass instead of both mass and granular pressure. Worthy of note, locomotor performance is repeatable, suggesting the history-dependent forces are a predictable function of body shape changes and granular remodeling. Future work elucidating such a predictive model could both facilitate design of more effective all-terrain robots and further understanding of the biology.

## Materials and methods

### Animal experiments

All experimental procedures carried out at Zoo Atlanta were conducted in accordance with the Georgia Institute of Technology IACUC protocol A14058. All procedures involving zoo animals were reviewed and approved by the Zoo Atlanta Scientific Research Committee.

*C. occipitalis* were collected by Kevin and April Young in the Colorado Desert near Yuma, Arizona, USA under scientific collection permits (SP790952, SP625775, SP666119) approved by the Arizona Game and Fish Department and held in the Physiological Research Laboratory at Georgia Tech. Neither the sex nor the age of the animals was determined; gender and age dependent

effects were beyond the scope of this study. All experimental procedures were conducted in accordance with the Georgia Institute of Technology IACUC protocols A14066 and A14067.

The temperature in the track way and snake holding area was measured prior to each trial. Lamps were used to ensure the temperature in both remained at $26 \pm 1$ °C , within the active range for *C. occipitalis* (*Klauber, 1939*). The heat lamps on the track way were turned off during data collection and LED lights were used for illumination.

Each day the *C. occipitalis* to be tested were transported from the housing facility to the lab where we conducted the trials. Snakes which were in the process of shedding were not used. During a trial, the snake was removed from its holding container and placed immediately in the fluidized track way. The snakes tended to be skittish and handling both during trials and in the housing facility was kept to a minimum. The animals would often immediately flee across the surface upon introduction to the track way; otherwise a light tail tap would elicit an escape response. If an individual did not respond to this stimulus they were returned to the holding container. If an individual failed to perform for three trials in a row they would be retired from the day's studies. Snakes were tested at most every other day with a maximum of two successful trials collected per day. A run was included if the snake performed at least four complete undulations moving along a straight trajectory at apparently constant speed.

## Calculation of waveform parameters

To measure $\xi$ we found the points of zero curvature; these corresponded to the inflection points of the waveform. We then calculated the arclength between these two points for the first half-wave on the body, multiplied by 2 to get the arclength of one full wave, $\lambda_s$, and divided the individual's length by the result.

We calculated $\theta$ of *C. occipitalis* using finite differences to estimate the x and y values for each segment's tangent vector. We generated sample serpenoid waveforms of known parameters and used these to determine that, in the presence of white Gaussian noise, the most accurate measurement of the tangent vector was obtained by subtracting the average position of the segment in question and three segments anterior from the average of the segment and the next three posterior segments. This method helped buffer against noise while still providing an accurate measurement. We found the maximum angle on the body at each time step and then averaged over all times in a trial in the to obtain $\theta_{max}$.

Calculating $\theta$ of the non-sand-specialists was challenging as in some cases direction of travel was difficult to determine or the waveform was tortuous. Thus, we estimated $\theta_m$ using $\lambda_s$ and the wave amplitude in curvature, $\kappa_m$, to determine the non-dimensional $\kappa_m \lambda_s$ (*Sharpe et al., 2015*). Assuming serpenoid curves, $\theta_m = \kappa_m \lambda_s / 2\pi$. We used finite differences to calculate $\kappa_m$ and the same procedure as described for $\theta$ to determine the best result was obtained by averaging over five points anterior and posterior to the point of inquiry.

## Bending beam model for snake segment motion

We model a snake body segment as a beam with midline length $\delta s$ and length of the inside of the curve $\delta s'$. For a given body radius, $r$, and radius of curvature, $R$, $\delta s' = \delta s \frac{R-r}{R}$. The speed the inside of the segment must change length from the unbent length of $\delta s$ to $\delta s'$ is $v_{shorten} = \frac{\delta s - \delta s'}{\delta t}$. We can write $\delta s - \delta s'$ as $\delta s (1 - \frac{R-r}{R})$. Using the relation $R = \kappa^{-1}$ we get $v_{shorten} = \delta s r \kappa$. Next, using $\kappa = \frac{d\theta}{ds}$ for a serpenoid curve $\theta(s,t) = \theta_m \sin(2\pi(\frac{\xi}{L}s + ft))$, we find the maximum curvature in terms of attack angle $\kappa_m = 2\pi \theta_m \frac{\xi}{L}$. Thus the maximum length change $\delta s - \delta s' = \delta s r 2\pi \theta_m \frac{\xi}{L}$. The time the segment has to undergo this change is a quarter of a period, so $\delta t = (4f)^{-1}$. Thus the shortening speed of the inside of a segment is

$$v = \delta s r 2\pi \theta_m \frac{\xi}{L} 4f.$$

Lastly, for the nominal values $f_o, \xi_o, \theta_{m,o}$ we get the ratio $\frac{v}{v_o} = \frac{\theta_m \xi f}{\theta_{m,o} \xi_o f_o}$ and the relative shortening speed is

$$\frac{v - v_o}{v_o} = \frac{\theta_m \xi f - \theta_{m,o} \xi_o f_o}{\theta_{m,o} \xi_o f_o}.$$

For a predetermined $v_{CoM}$, MATLAB's fsolve is used to find the undulation frequency for a given $\theta_m$ and $\xi$ which results in no-slip motion at that value of $v_{CoM}$.

## Dissection and torque estimation

Four adult *C. occipitalis* (SVL 33.0 ± 3.7 cm, mass 16.4 ± 3.4 g) which had preserved in formalin and stored in ethanol were scalened and the body (from the posterior margin of the quadrates to the cloaca) was cut into 10 equal length segments, which were then weighed intact. Segments were then eviscerated and the epaxial muscle mass (consisting of the largest muscles, the m. multifidus, m. semispinalis-spinalis, m.longissimus dorsi, and m. iliocostalis *Jayne, 1988*) was removed, and the viscera, muscle, and remaining body tissue were weighed. All tissues were kept moist in 70% ethanol and dabbed dry before weighing. Snakes were an average of 22.1 ± 3.8% muscle by mass, but this proportion varied regionally due to uneven total segment masses and viscera masses; absolute muscle mass was highest at midbody and decreased anteriorly and posteriorly, though the highest muscle mass was only 27.0 ± 3.8% higher than the average.

Because of postmortem and preservation distortion of body shape (facilitated by the mobile ribs of snakes), average body radius was computed from SVL and mass by treating the snake as a uniform cylinder with a tissue density of 1.05 g cm$^3$ (typical for vertebrate tissues). As the muscular lever arms for lateral flexion are unknown for any snake, we estimated the maximum lever arm as 1/2 the radius of the body; while some epaxial muscles (e.g., m. iliocostalis) may have larger lever arms, others likely have much lower lever arms (m. multifidus and semispinalis-spinalis). Similarly, muscular PSCA for the entire epaxial muscle group was computed as a cylinder based on SVL and total muscle mass; while snake epaxial musculature is highly complex, none of the muscles show strong pennation. Peak isometric muscle force was estimated based on the standard 30 N cm$^2$ value seen in most vertebrate muscles, and divided by two to account for unilateral activation (*Jayne, 1988*). Although the maximal shortening speed and shape of the force-velocity relationship is unknown in snakes, we assumed that during lateral undulation, snakes would be operating near their peak isotonic power, and thus with a force of half the peak isometric muscle force; as activation/deactivation kinetics and length tension properties are also unknown in snakes, we did not attempt to account for these. Peak torque was computed as this force divided by estimated lever arms. Although many crucial properties are unknown, this value represents a charitably high estimate of peak torque; this value would be depressed by steeper force-velocity curves, departure from the plateau of the length-tension curve, incomplete activation during the work loop, or lower muscle lever arms, while the higher muscle mass at midbody and slightly larger vertebrae would increase the peak torque.

## Acknowledgements

We thank Kevin and April Young for collecting the *C. occipitalis* and Mark Lowder for his assistance in creating the non-sand-specialist tracking code. This work was supported by NSF PoLS PHY-1205878, PHY-1150760, and CMMI-1361778. ARO W911NF-11-1-0514, ARO W911NF-18-1-0120, ARO W911NF-18-1-0118 (KK and SK), NDSEG 32 CFR 168a (PES), and the Simons Southeast Center for Mathematics and Biology.

## Additional information

### Funding

| Funder | Grant reference number | Author |
| --- | --- | --- |
| National Science Foundation | PHY1205878 | Perrin E Schiebel<br>Henry C Astley<br>Jennifer M Rieser<br>Alex M Hubbard<br>Daniel I Goldman |

| Army Research Office | W911NF-11-1-0514 | Jennifer M Rieser<br>Perrin E Schiebel<br>Henry C Astley |
|---|---|---|
| American Society for Engineering Education | 32 CFR 168a | Perrin E Schiebel |
| Army Research Office | W911NF-18-1-0120 | Perrin E Schiebel<br>Henry C Astley<br>Jennifer M Rieser<br>Christian Hubicki<br>Alex M Hubbard<br>Kelimar Diaz<br>Daniel I Goldman |
| Simons Foundation | | Perrin E Schiebel<br>Kelimar Diaz<br>Daniel I Goldman |
| National Science Foundation | PHY-1150760 | Perrin E Schiebel<br>Jennifer M Rieser<br>Henry C Astley<br>Alex M Hubbard<br>Daniel I Goldman |
| National Science Foundation | CMMI-1361778 | Perrin E Schiebel<br>Jennifer M Rieser<br>Henry C Astley<br>Daniel I Goldman |
| Army Research Office | W911NF-18-1-0118 | Ken Kamrin<br>Shashank Agarwal |

The funders had no role in study design, data collection and interpretation, or the decision to submit the work for publication.

## Author contributions
Perrin E Schiebel, Conceptualization, Data curation, Software, Formal analysis, Validation, Investigation, Visualization, Methodology; Henry C Astley, Conceptualization, Data curation, Software, Formal analysis, Supervision, Validation, Investigation, Methodology; Jennifer M Rieser, Conceptualization, Data curation, Software, Supervision, Validation, Investigation; Shashank Agarwal, Software, Formal analysis, Methodology; Christian Hubicki, Data curation, Formal analysis, Validation, Investigation; Alex M Hubbard, Kelimar Diaz, Formal analysis, Investigation; Joseph R Mendelson III, Conceptualization, Resources, Supervision, Funding acquisition, Project administration; Ken Kamrin, Resources, Supervision, Project administration; Daniel I Goldman, Conceptualization, Resources, Funding acquisition, Project administration

## Author ORCIDs
Perrin E Schiebel (iD) https://orcid.org/0000-0003-2424-829X
Daniel I Goldman (iD) https://orcid.org/0000-0002-6954-9857

## Ethics
Animal experimentation: This study was performed in strict accordance with the recommendations in the Guide for the Care and Use of Laboratory Animals of the National Institutes of Health. All experimental procedures using the non-sand-specialists were conducted in accordance with the Georgia Institute of Technology institutional animal care and use committee (IACUC) protocol A14058. All procedures involving these zoo animals were reviewed and approved by the Zoo Atlanta Scientific Research Committee. All experimental procedures using C. occipitalis were conducted in accordance with the Georgia Institute of Technology IACUC protocols A14066 and A14067. All experiments were non-invasive and every effort was made to minimize distress.

## Decision letter and Author response
Decision letter https://doi.org/10.7554/eLife.51412.sa1
Author response https://doi.org/10.7554/eLife.51412.sa2

## Additional files

### Supplementary files
• Transparent reporting form

### Data availability

Tracked snake midlines and granular drag data have been deposited to SMARTech online repository. Codes are available on GitHub at https://github.com/PerrinESchiebel/SnakeTrackingCodes (copy archived at https://github.com/elifesciences-publications/SnakeTrackingCodes).

The following dataset was generated:

| Author(s) | Year | Dataset title | Dataset URL | Database and Identifier |
|---|---|---|---|---|
| Schiebel PE, Astley HC, Rieser JM, Agarwall S, Hubicki C, Hubbard AM, Cruz K, Mendelson J, Kamrin K, Goldman DI | 2019 | Mitigating memory effects during undulatory locomotion on hysteretic materials dataset | https://smartech.gatech.edu/handle/1853/61852 | SMARTech, 61852 |

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
