## [Decision Letter]

**Acceptance summary:**

This work addresses how animals like snakes deal with the deformations they leave behind on the surface of deformable materials (sand, mud) on which they slither. Insights from studies of a variety of snakes and a "robophysical" model in which a slender plate is dragged through sand are combined with theoretical ideas from granular physics and the fluid mechanics of locomotion by slender objects to obtain a clear picture of the energetic tradeoffs necessary for efficient locomotion.

**Decision letter after peer review:**

Thank you for submitting your article "Mitigating memory effects during undulatory locomotion on hysteretic materials" for consideration by *eLife*. Your article has been reviewed by three peer reviewers, and the evaluation has been overseen by a Reviewing Editor and Andrew King as the Senior Editor. The following individuals involved in review of your submission have agreed to reveal their identity: Michael J Shelley (Reviewer #2); Stephen Morris (Reviewer #3).

The reviewers have discussed the reviews with one another and the Reviewing Editor has drafted this decision to help you prepare a revised submission.

Summary:

This paper presents an ambitious and rather comprehensive study of the locomotion of a wide selection of snake species across the surface of a granular medium. To my knowledge, such a broad study has not been attempted before. The main strength of the work is the broad and interdisciplinary approach, combining quantitative measurements, theory, connections to snake anatomy, field observations and robot models. As the authors point out, there is a very well-developed theory for propulsion in Newtonian fluids (like water), where resistive force theory (RFT) serves as a good approximation to the full hydrodynamic problem of viscous flow around slender objects. Here the question is what kind of local theory analogous to RFT might hold when the substrate on which the animal moves is permanently deformed by the motion. The physical model is convincingly validated with simple plate dragging experiments. The paper contains a wealth of interlocking details and observations, including identifying when and why the locomotion strategy fails for some non-adapted species. It provides a clear and insightful picture of snake locomotion that might be used to advance robotic analogs. In summary, this paper is a very good, interesting, and thorough study of a mostly unexplored topic of broad interest.

Essential revisions:

1) The paper presents a very detailed and comprehensive study, but one that is scattered over too many disconnected pieces, making the narrative difficult to follow. There is a Main Text, very detailed and compact figures with long captions, a huge array of appendices and supplementary information, and appended videos. The effect is to diffuse the information rather widely, making reading difficult. There is overlap and repetition. We have not previously encountered a relatively short paper with 9 appendices. The authors must reformat the paper to make use of figure supplements in order that the logic and results are more clearly spelled out.

Moreover, the authors cite the supplementary material as if it were a separate publication. It is thus cited differently than Appendix material! We found this quite confusing.

Some minor details of method and snake husbandry might be usefully combined into one Materials and methods appendix with short subsections, while other appendices and supplemental material should be put back into the Main Text. We consider the mechanical RFT model to be a main contribution of the paper and thus should be mostly found in the Main Text. The Introduction and conclusions need to provide a clearer roadmap to all the interlocking parts of the argument and results.

2) There is also a distinct lack of detail presented on the mathematical/physical modeling. A good example of this lack of detail is found in the section on the drag model. We found no detail in the paper about the form of the "static stress" σ_0_.

In addition to providing those details in the paper, we would suggest that the authors provide (at least) some heuristic arguments explaining the typical scale of surface stresses they obtain (say, 0.1 N/cm^2^). Surely this can be explained in terms of grain size, coefficient of static friction, gravity, etc. Without this, the reader has learned nothing from the fact that an unspecified model is consistent with the data.

3) The issue of the drag anisotropy is a fascinating one, particularly in comparison to the behavior of long slender objects in low Reynolds number Newtonian fluids. But here we were slightly confused about the results, and think the presentation needs to be reworked. Consider the following: in normal slender body hydrodynamics there are drag coefficients for motion normal (perpendicular) and tangential (parallel) to the filament, call them ζ_para_ and ζ_perp_, and one writes the total drag force F (per unit length) on a filament with velocity v as

F = (ζ_para_tt + ζ_perp_nn)dotv

where ζ_perp_ approximately 2 ζ_para_ = 4 pi eta /(log(L/d)+const), where η is the viscosity, L the total length, and d the filament diameter, and the constant is order unity. The coefficients zeta (which are basically the C_n_ and C_t_ coefficients in subsection “Drag anisotropy is not strongly dependent on speed or depth”) thus do not depend on the angle of motion. So, we think the real question here is whether the experimental results in this paper are consistent with such constant zetas. Therefore, instead of (or in addition to) plotting the ratio σ_n_/σ_t_ as a function of β_d_ in Figure 6A, consider plotting versus tan(β_d_) and/or a plot of (σ_n_/σ_t_)/tan(β_d_) vs β_d_ to see if a straight line emerges, which would be an indication of fluid-like behavior (or deviation from it). Also, the quantity K_fluid_ mentioned in the figure appears to be undefined.

4) Continuing with this point, the authors need to write down precisely a mathematical statement of resistive force theory for these systems. Is it like what is written above except for the definitions of the zetas (Cs), or something else?

5) The section comprising paragraph four-six in subsection “Surface resistive force theory model” is similarly in need of revision. It continually refers to supplemental material and simply says whether the model works or doesn’t but does not convey any real physical insight to the reader without turning to that material.

6) The authors may wish to note that another medium that has memory is a fluid moving at high Reynolds number. Recent work from the lab at Courant has shown that hysteresis arises in model experiments of flocks due to the finite lifetime of vortices shed by leading swimmers.

7) The authors frequently make reference and comparisons to subsurface locomotion and to low-Re swimmers, without really explaining what Reynolds number is. They seem to assume the reader is simply familiar with these cases, without explanation. It might be useful to briefly bring together these comparisons in the Introduction, so that later reference to them seem less disconnected.

---

## [Author Response]

Essential revisions:1) The paper presents a very detailed and comprehensive study, but one that is scattered over too many disconnected pieces, making the narrative difficult to follow. There is a Main Text, very detailed and compact figures with long captions, a huge array of appendices and supplementary information, and appended videos. The effect is to diffuse the information rather widely, making reading difficult. There is overlap and repetition. We have not previously encountered a relatively short paper with 9 appendices. The authors must reformat the paper to make use of figure supplements in order that the logic and results are more clearly spelled out.Moreover, the authors cite the supplementary material as if it were a separate publication. It is thus cited differently than Appendix material! We found this quite confusing.Some minor details of method and snake husbandry might be usefully combined into one Materials and methods appendix with short subsections, while other appendices and supplemental material should be put back into the Main Text. We consider the mechanical RFT model to be a main contribution of the paper and thus should be mostly found in the Main Text. The Introduction and conclusions need to provide a clearer roadmap to all the interlocking parts of the argument and results.

We thank the reviewers for drawing our attention to these weaknesses in our paper. Please find below an itemized list of changes we made to aid reader comprehension.

– As per the reviewers’ suggestion we combined non-essential animal experimental information (collection and IACUC protocols, animal handling procedures) into a single appendix. Information about the animal experiment setup (bed size, cameras, and tracking procedure) is moved to the main text, subsection “Kinematics and ability vary among species.”. Description of *C. occipitalis* husbandry and the materials used in the construction of the fluidized bed used in the specialist experiments is in the supplemental information as this information is not pertinent to the results nor of broad interest.

– We kept an appendix detailing the calculation of the waveform parameters *θ* and *ξ* since this was carried out differently for specialists and nonspecialists, but the details of the procedure are not necessary to understand the results. Notes on this calculation that were in the supplemental were condensed and combined with the Appendix.

– Description of the granular drag experimental setup is moved from the Appendices to the main text, subsection “Granular drag measurements.”.

– Most of the details pertaining to RFT have been moved from the Appendix and supplemental to the main text. The only RFT information remaining in the supplemental is an explanation of the calculation of mechanical cost-of-transport and bodylengths per undulation cycle. We briefly mention these measurements in the main text as they are commonly used in the animal literature such that some readers may be interested in the prediction. However, as these results do not have any bearing on the rest of the study they are presented in the supplemental information. We added a supplemental table of the fourier-fit coefficients used to model the parallel forces.

– The Appendices on the measurements of body lifting and laser line reconstruction of the granular surface have been moved to supplemental. Both experiments and their results support observations and modeling decisions in the main text, however, they are not necessary to understanding the paper or results and we feel it would distract the reader to include this detail in the main text.

– We removed all references to the supplemental material as a citation and replaced them with figure supplements.

The Introduction and conclusions need to provide a clearer roadmap to all the interlocking parts of the argument and results.

We added some more detail on the differences between low-Re swimming, high-Re swimming, and fluids versus granular materials to the Introduction to better set up the main ideas in the paper (changes are further detailed below in response to subsequent critiques). We expanded the overview of the paper contents at the end of the Introduction to better orient the reader to the logical flow. Throughout the paper we edited and in places expanded the transitions between sections to better link them and provide more guidance to the reader.

2) There is also a distinct lack of detail presented on the mathematical/physical modeling. A good example of this lack of detail is found in the section on the drag model. We found no detail in the paper about the form of the "static stress" σ0.In addition to providing those details in the paper, we would suggest that the authors provide (at least) some heuristic arguments explaining the typical scale of surface stresses they obtain (say, 0.1 N/cm2). Surely this can be explained in terms of grain size, coefficient of static friction, gravity, etc. Without this, the reader has learned nothing from the fact that an unspecified model is consistent with the data.

We thank the reviewers for this suggestion. We note that the original text did explain the method for obtaining *σ_o_*, however, our explanation was clearly lacking clarity. As such we substantially reworked this section to be more explicit. We included the model in equation form, and added an estimation of the static stress using hydrostatic pressure, included as follows,

“The animal behavior indicated that friction between the body and the GM and local, dissipative interactions within the GM were the dominant forces in the system. […] *σ_o_*was calculated by subtracting *ρv_d_*^2^ from *σ_n_*measured at the three lowest speeds collected at a given *z* and averaging the result.”

3) The issue of the drag anisotropy is a fascinating one, particularly in comparison to the behavior of long slender objects in low Reynolds number Newtonian fluids. But here we were slightly confused about the results, and think the presentation needs to be reworked. Consider the following: in normal slender body hydrodynamics there are drag coefficients for motion normal (perpendicular) and tangential (parallel) to the filament, call them ζpara and ζperp, and one writes the total drag force F (per unit length) on a filament with velocity v asF = (ζparatt + ζperpnn)dotvwhere ζperp approximately 2 ζpara = 4 pi eta /(log(L/d)+const), where η is the viscosity, L the total length, and d the filament diameter, and the constant is order unity. The coefficients zeta (which are basically the Cn and Ct coefficients in subsection “Drag anisotropy is not strongly dependent on speed or depth”) thus do not depend on the angle of motion. So, we think the real question here is whether the experimental results in this paper are consistent with such constant zetas. Therefore, instead of (or in addition to) plotting the ratio σn/ σt as a function of βd in Figure 6A, consider plotting versus tan(βd) and/or a plot of (σn/ σt)/tan(βd) vs βd to see if a straight line emerges, which would be an indication of fluid-like behavior (or deviation from it). Also, the quantity Kfluid mentioned in the figure appears to be undefined.

We thank the reviewers for pointing out this lack of clarity and the suggestion of plotting the ratio in relation to tan(*β_d_*) to highlight the difference between drag in GM versus viscous fluids. We updated the first paragraph in subsection “Drag anisotropy is not strongly dependent on speed or depth.” to point out that it is known in subsurface drag that the prefactors are functions of drag angle.

We included the following paragraph in subsection “Drag anisotropy is not strongly dependent on speed or depth.”, which references the plot of σnσttan(βd) which has been added to Figure 6B.

“In viscous fluids the constants *C_n_*and *C_t_*are independent of drag angle such that σnσttan(βd) is constant. Consistent with results for subsurface drag, Maladen et al., 2009, 2011b, we find that at the surface σnσttan(βd) is a nonlinear function of *β_d_*(Figure 6B).”

4) Continuing with this point, the authors need to write down precisely a mathematical statement of resistive force theory for these systems. Is it like what is written above except for the definitions of the zetas (Cs), or something else?

We expanded subsection “Surface resistive force theory model.“ to include the mathematical definition of RFT used in our model. We moved most of the description of the calculation from the Appendix and supplemental to the main text with the exceptions described in response to revision 1.

Indeed, the formulation of granular RFT is as written above with functions *f_n_*and *f_t_*in place of the constants. In our case, we note that we used a fourier fit to the data to obtain *f_t_*(as noted in the main text). We chose to do this to allow us to carry out the RFT calculation without having a model for the form of the tangential forces as the models previously used in subsurface drag did not sufficiently capture the shape of the curve.

5) The section comprising paragraph four-six in subsection “Surface resistive force theory model” is similarly in need of revision. It continually refers to supplemental material [21] and simply says whether the model works or doesn’t but does not convey any real physical insight to the reader without turning to that material.

We thank the reviewers for pointing out this lack of clarity. To address this, we moved the figure comparing RFT-calculated movement speeds with those measured in experiment from the supplemental to the main text (Figure 6 in the revised manuscript). We expanded our Discussion of this comparison in the main text to better present the important findings.

6) The authors may wish to note that another medium that has memory is a fluid moving at high Reynolds number. Recent work from the lab at Courant has shown that hysteresis arises in model experiments of flocks due to the finite lifetime of vortices shed by leading swimmers.

This is an excellent point. We have added the following to the Introduction to include the time-dependent flows occurring at high-Re:

“At the other end of the spectrum, animals move in materials whose state depends on the history of interaction. […] However, unlike fluids, they can bear internal stress. At the free surface, gravity is often insufficient to return mounds of material created by a locomotor to the undisturbed state, exemplified by the tracks left behind by a passing animal.”

7) The authors frequently make reference and comparisons to subsurface locomotion and to low-Re swimmers, without really explaining what Reynolds number is. They seem to assume the reader is simply familiar with these cases, without explanation. It might be useful to briefly bring together these comparisons in the Introduction, so that later reference to them seem less disconnected.

We thank the reviewers for pointing out this oversight. We have added a definition of Reynolds number to the Introduction as follows:

“At one end of the spectrum, where deformations are short-lived Cohen and Boyle, 2010, small fluid swimmers like the nematode *Caenorhabditis elegans* Wen et al., 2012, spermatazoa Gray, 1953, and bacteria in water Rodenborn et al., 2013, are propelled by the viscous force of the material resisting the animal’s +body shape changes. […] Notably, in both viscous and frictional fluids, the material surrounding the submerged swimmers continuously re-flows around the body of the animal such that the animal is always surrounded by material.”

In addition to the added the paragraph discussing high-Re swimming (copied above in response to comment 6) this provides a better explanation for motion in viscosity versus inertia-dominated systems.